# Locations and structures of influenza A virus packaging-associated signals and other functional elements via an *in silico* pipeline for predicting constrained features in RNA viruses

**Emma Beniston, Jordan P. Skittrall**⬤*

Department of Pathology, University of Cambridge, Cambridge, United Kingdom

* jps55@cam.ac.uk

**Data Availability Statement:** There are no primary data in the paper. Data were acquired from the GISAID database (https://gisaid.org/). Lists of

## Abstract

Influenza A virus contains regions of its segmented genome associated with ability to package the segments into virions, but many such regions are poorly characterised. We provide detailed predictions of the key locations within these packaging-associated regions, and their structures, by applying a recently-improved pipeline for delineating constrained regions in RNA viruses and applying structural prediction algorithms. We find and characterise other known constrained regions within influenza A genomes, including the region associated with the PA-X frameshift, regions associated with alternative splicing, and constraint around the initiation motif for a truncated PB1 protein, PB1-N92, associated with avian viruses. We further predict the presence of constrained regions that have not previously been described. The extra characterisation our work provides allows investigation of these key regions for drug target potential, and points towards determinants of packaging compatibility between segments.

## Author summary

Influenza A viruses infect birds and mammals, including humans. Their genetic code is contained in multiple segments; all these segments need to be packaged inside virus particles for effective transmission. Mixing of segments between viruses that infect different organisms can result in pandemics. We have developed a computational pipeline for finding regions within the genetic code where mutation is less often seen; these regions are often important for virus lifecycle. In the case of influenza A, many of the regions we identify are associated with ensuring the segments are efficiently packaged. Other regions are likely to be important in allowing the virus to optimise the ratios of the different proteins it produces. Our work guides how to investigate these regions further for drug target potential.

accession numbers of sequences used in the study are provided in S18–S112 Tables. The code used to analyse the data has been published previously, and is available at https://github.com/skittrall/RNAdescent.

**Funding:** EB was funded by the Cambridge Mathematics of Information in Healthcare Hub at the Centre for Mathematical Sciences, University of Cambridge (https://www.cmih.maths.cam.ac.uk/), and by the H.E. Durham Fund, King's College Cambridge (https://www.kings.cam.ac.uk/). JPS was funded by the NIHR Clinical Lecturer programme (https://www.nihr.ac.uk/). The funders played no role in the study design, data collection and analysis, decision to publish, or preparation of the manuscript.

**Competing interests:** The authors have declared that no competing interests exist.

# Introduction

Influenza illness remains a significant cause of human mortality and morbidity worldwide, with influenza A virus the largest contributor to this burden. Central estimates of annual worldwide seasonal influenza-associated mortality exceed 100,000, and actual mortality may be substantially higher; annual hospitalisations are around ten times this number [1, 2]. There were four influenza A pandemics in the twentieth century, and were a pandemic of similar magnitude to the 1918–19 pandemic to happen today, it is estimated that mortality would be in the tens of millions [3, 4]. Outbreaks of avian influenza in poultry can lead to culling of up to millions of birds and economic costs up to billions of dollars [5, 6]. Consequently, there remains a strong imperative to optimise influenza vaccines and develop further antivirals, as well as to develop diagnostics that are minimally susceptible to mutations yet able to detect reassortment events, and to understand molecular mechanisms that may lead to pandemics.

Influenza A is known to have multiple features requiring constraint at RNA level beyond that required for amino acid conservation, including overlapping open reading frames, frameshift and splice sites, and structured regions. In addition, the segmented nature of its genome imposes particular requirements upon the virus in recognising and packaging its viral RNA (vRNA). The area of influenza A packaging has been heavily studied (reviewed, for example, in references [7, 8]): current understanding is that the virus usually packages exactly one of each segment per virion, most likely using a set of signals in the vRNA that interact with each other and with the nucleoprotein in ways that are not yet fully specified, and possibly with some redundancy meaning that mutations in packaging signals may be less detrimental to packaging than might otherwise be anticipated. Redundancy may mean that some packaging interactions are more prominent in some influenza A subtypes than others. This is important when considering reassortment of segments between subtypes and hence generation of pandemic strains, because it raises the possibility that molecular constraints mean some reassortants are more likely to occur than others. Understanding packaging signals may therefore focus epidemiological and public health efforts to reduce pandemic risk, by redirecting surveillance and risk mitigation away from strains less able to reassort to cause pandemics, and towards strains more able to reassort to cause pandemics. More generally, identifying and characterising constraint in RNA may improve vaccine strain design, improve PCR primer design, and ascertain candidate drug targets.

Computational methods developed to detect unusually high RNA-level constraint, corresponding to selective pressure to maintain an important lifecycle feature, have previously characterised important motifs in influenza A. Identifying such motifs has led to the characterisation of packaging signals and of frameshifting to an alternative open reading frame [9, 10]. Since the application of those methods, much larger datasets of influenza A genomic data have become available. Methodological refinements have also since become available, aimed at better delineation of constrained regions of RNA and improved signal-to-noise discrimination [11, 12]. Given these improvements in datasets and available analysis techniques, and the ongoing clinical and public health importance of influenza A, we have undertaken an updated analysis.

## Materials and methods

### Viral sequences

Influenza A virus sequences were downloaded from the GISAID database [13]. Viruses were selected with the haemagglutinin (H, HA)/neuraminidase (N, NA)/host type combinations of H1N1/human, H1N2/swine, H3N2/human, H5N1/avian, H5N8/avian, H7N7/avian, H7N9/

avian and H7N9/human. Sequences with submission dates between 2008-01-01 and 2022-07-01 were selected. Sequences dating from the 2009 pandemic and later were chosen for the H1N1 subtype (i.e. H1N1 sequences were H1N1pdm09 lineages). To ensure that, as far as possible, the dataset reflected sequences from natural infections, viruses annotated as having undergone multiple passages *ex vivo* were excluded. Where the database recorded multiple submissions arising from the same initial isolate, only one submission was retained in the dataset; the submission reflecting the initial isolate was retained where available. Nucleotide sequences for the PB2, PB1, PB1-F2, PA, PA-X, HA, NP, NA, M1, M2, NS1 and NS2 genes were extracted. Genes whose sequencing was incomplete or ambiguous, or that contained frameshift mutations leading to premature termination, were excluded. S1 Table details the number of sequences analysed for each gene/subtype/host combination. A full listing of the accession numbers of sequences analysed may be found in S18–S12 Tables.

## Sequence analysis

Sequence analysis to find regions of increased constraint was carried out separately for each HA/NA/host type combination, and within each combination for each gene. Sequences were aligned at amino acid level using MAFFT [14] with options selecting the FFT-NS-1 method and not using the FFT approximation in group-to-group alignment. Alignments were back-translated (using the original sequence data) to nucleotide alignments with codon-level alignment.

Analysis was carried out in Mathematica [15] using RNAdescent algorithms [9, 11, 12, 16]. To summarize: each codon position in each alignment was allocated a score, corresponding to the sum of the Hamming distances between the nucleotides forming the codon for each possible pairing of sequences from the alignment, divided by the sum of the Hamming distances that would have arisen had the codon usage frequency equalled the overall observed codon usage frequency for the amino acids seen at the locus. (The codon usage frequency for all twelve genes analysed—eleven for H1N1pdm09, which has a non-functional PB1-F2 gene—was taken into account, including overlapping open reading frames, but not including the portion of PA-X prior to the frameshift.) Gap characters were not taken to contribute to Hamming distances, but any locus with a gap frequency exceeding 10% was excluded from analysis. Each locus was also allocated a weighting, corresponding to the Shannon information of the observed codon usage. In the first analysis, the weighted score was used as the vertical deviation in a random walk-like process, and the weights used as the horizontal deviation. The most extreme negative $Z$-statistic for the walk was calculated, and the region corresponding to the statistic deemed significantly constrained if the statistic was significant at the 5% level by bootstrap rearrangement of loci. If a region was deemed significantly constrained, then the process was repeated for the remainder of the gene (with the significant region excised). To account for possible interference between multiple significant regions, where a non-significant region was found, then the analysis was repeated with the region corresponding to the next most extreme $Z$-statistic excised; regions found to be significant only after re-analysis for interfering signals were marked as such. The second analysis was similar to the first analysis, save that the scores of the first analysis were replaced by ranks (with the lowest rank given to the locus with the lowest score, etc.), and then analysis performed as before but on the ranks instead of their associated scores. We have shown previously that under some circumstances, the analysis of ranked scores may improve detection of truly significant regions [12], but as there is sometimes a specificity cost to using this method we performed analyses using both raw and ranked scores and then compared the results.

Alignments of regions deemed significantly constrained by the above analysis were used as input to RNAalifold [17, 18]. Both coding (+ sense) and genomic (− sense) RNA (cRNA/

vRNA) was evaluated. Where the consensus sequence at a position in the alignment was a gap character, this was removed from the RNAalifold analysis. RNAalifold was run with ribosum scoring enabled, lonely pairs disallowed, and G-quadruplex formation enabled. RNAalifold is a minimum free energy RNA folding prediction program, which uses covariation data from sequence alignments to modify predicted base-pairing.

In some cases (primarily for subtypes with human hosts), the number of sequences used as input to RNAalifold resulted in a memory requirement exceeding that of a desktop computer; in these cases, a single representative of each primary structure seen within the region was selected and fold prediction undertaken with this reduced set of sequences.

Regions deemed significantly constrained were inspected for out-of-frame coding potential. Existing literature was reviewed for explanations for observed sequence constraint.

## Results

A summary of all the constrained regions found by our analysis can be found in Table 1. A detailed description of the exact extent of each region found in each analysis, plus statistics from the analysis output, may be found in S2–S17 Tables. Codon-by-codon output with the results of each constraint analysis, plus sequences highlighted to display constrained regions and plots showing constraint information, are supplied for each subtype/host combination as Mathematica files and as pdf files, in S1–S8 Code.

### Characterisation of packaging-associated regions

Similar to previous analyses based on sequence constraint [9, 11], many of the constrained regions we find consistently across the different subtypes analysed are in regions associated with packaging. However, because our analysis uses updated methods on larger datasets, we are able to delineate more closely the boundaries of the constrained regions and use these delineated regions as the basis for structural prediction. We can therefore make computational predictions of secondary RNA structures formed by constrained regions, and can review structural predictions of conserved regions for complementary regions potentially involved in packaging. For ease of reference, we describe our findings in canonical order through the genome (i.e. in segment order, 5′ to 3′ in cRNA). Regions where the explanations for constraint are less apparent are discussed in S1 Appendix.

**Two highly conserved stem-loops in PB2 3′ vRNA.** Structural prediction using RNAalifold on the 3′ vRNA (corresponding to 5′ cRNA) regions deemed constrained in PB2 by RNA-descent predicts two well-conserved stem-loops (Fig 1). (Loops correspond to codons 16–17 and 27–28 of PB2, equivalently NC_007373.1 nucleotides 73–78 and 106–111.) Inspection of the per-codon output shows that the codons in the stem immediately flanking the loop are amongst the most constrained individual codons in all subtypes analysed, and that they are more constrained than either the codons predicted to form the loop or the codons further from the loop. The codon-level constraint results provide strong corroboration of the stem-loop predictions.

In coding (+) sense, the consensus sequence for PB2 codons 15–18 across all eight subtype/host combinations studied is CGC_ACU_CGC_GAG. The maintenance of multiple CpGs in both + and—sense in this region provides additional corroboration that there is strong selection to maintain this sequence. (We note codon usage is one of the factors our algorithm uses to determine constraint, so the presence of these CpGs forms part of the evidence used to determine the region to be constrained.) This region has also been predicted to be structured in coding sense by experiments using DMS-MaPseq [21], and our coding sense predictions have similar structure (S1–S8 Figs).

**Table 1. Summary of constrained regions found in influenza types, by analysis method.**

| Description of constrained region | Human hosts | | | | | | Swine | | Avian hosts | | | | | | | |
|---|---|---|---|---|---|---|---|---|---|---|---|---|---|---|---|---|
| | H1N1 | | H3N2 | | H7N9 | | H1N2 | | H5N1 | | H5N8 | | H7N7 | | H7N9 | |
| | raw | rnk | raw | rnk | raw | rnk | raw | rnk | raw | rnk | raw | rnk | raw | rnk | raw | rnk |
| PB2 5′ packaging-associated [19–21] | | ✓ | | | | ✓ | ✓ | ✓ | ✓ | ✓ | ✓ | ✓ | ✓ | ✓ | ✓ | ✓ |
| PB2 unknown constrained region ca. nt900–1110 | | | | | | | | | ✓ | | | | ✓ | | | |
| PB2 3′ packaging-associated [9, 19, 22–27] | | ✓ | ✓ | ✓ | ✓ | ✓ | ✓ | ✓ | ✓ | ✓ | ✓ | ✓ | ✓ | ✓ | ✓ | ✓ |
| PB1 5′ packaging-associated [19, 20, 24] | | | | | | | | | ✓ | | | | | | | ✓ |
| PB1-F2 initiation region | | | | | | | | | | | | | ✓ | | | |
| PB1-N40 initiation region | | | | | | | | | | | | | ✓ | | | |
| PB1-N92 initiation region | | | | | | | | | ✓ | | ✓ | | ✓ | ✓ | | ✓ |
| PB1 3′ packaging-associated [19–21, 24, 25, 27] | | ✓ | ✓ | ✓ | ✓ | ✓ | ✓ | ✓ | ✓ | ✓ | ✓ | ✓ | ✓ | ✓ | ✓ | ✓ |
| PA 5′ packaging-associated [20, 24] | | | | ✓ | | | ✓ | | ✓ | | | | ✓ | ✓ | ✓ | ✓ |
| PA unknown constrained region ca. nt400 | | | | ✓ | | | | | | | | | | | | ✓ |
| PA-X proposed frameshift stimulator | | ✓ | ✓ | ✓ | ✓ | ✓ | ✓ | ✓ | ✓ | ✓ | ✓ | ✓ | ✓ | ✓ | ✓ | ✓ |
| PA/PA-X frameshifted region [10] | | ✓ | ✓ | ✓ | ✓ | ✓ | ✓ | ✓ | ✓ | ✓ | ✓ | ✓ | ✓ | ✓ | ✓ | ✓ |
| PA unknown constrained region pre-nt1300 | | ✓ | | | | | | | | | | | | | | |
| PA unknown constrained region ca. nt1800 | | | | ✓ | | | | | ✓ | | | | | | ✓ | |
| PA 3′ packaging-associated (often long region) [19, 24, 25] | | ✓ | ✓ | ✓ | ✓ | ✓ | ✓ | ✓ | ✓ | ✓ | ✓ | ✓ | ✓ | ✓ | ✓ | ✓ |
| HA unknown constrained region ca. nt800 | | | | | | | | | | | | ✓ | | | | |
| HA conserved stem-loops near cleavage site ca. nt1000 [28–30] | | | | | | ✓ | | | | | | | | | | |
| HA 3′ packaging-associated [31–33] | | ✓ | ✓ | ✓ | ✓ | ✓ | ✓ | ✓ | ✓ | | ✓ | ✓ | ✓ | ✓ | ✓ | ✓ |
| NP 5′ packaging-associated [27, 34, 35] | | | | | | ✓ | ✓ | ✓ | ✓ | ✓ | ✓ | ✓ | ✓ | ✓ | ✓ | ✓ |
| NP possible artefact ca. nt700–1200/apparently conserved region ca. nt1200 | | | | | | | | | ✓ | | ✓ | | | | | |
| NP 3′ packaging-associated [27, 34–37] | | | | ✓ | | ✓ | ✓ | ✓ | ✓ | ✓ | ✓ | ✓ | ✓ | ✓ | ✓ | ✓ |
| NA unknown constrained region ca. nt300–500 | | | | | | | | | | | ✓ | ✓ | | | ✓ | ✓ |
| NA unknown constrained region ca. nt600 | | | | ✓ | | | ✓ | | | | | | | | | |
| NA unknown constrained region post-nt800 | | | | | | | | | ✓ | | | ✓ | | | | |
| NA unknown constrained region ca. nt1000 [27] | | | | | | | | | ✓ | | | ✓ | | | | |
| NA 3′ packaging-associated [9, 37–39] | | ✓ | | ✓ | | | ✓ | ✓ | ✓ | ✓ | ✓ | ✓ | ✓ | ✓ | ✓ | ✓ |
| M2 splice donor | | | ✓ | ✓ | ✓ | ✓ | ✓ | ✓ | ✓ | ✓ | ✓ | ✓ | ✓ | ✓ | ✓ | ✓ |
| M 5′ packaging-associated [40, 41] | ✓ | ✓ | ✓ | ✓ | ✓ | ✓ | ✓ | ✓ | ✓ | ✓ | ✓ | ✓ | ✓ | ✓ | ✓ | ✓ |
| M42 ORF overlap [42] | ✓ | | ✓ | ✓ | ✓ | | | | | | ✓ | ✓ | ✓ | ✓ | ✓ | ✓ |
| M4 splice junction [42] | ✓ | | ✓ | ✓ | ✓ | | | | ✓ | | ✓ | ✓ | ✓ | ✓ | ✓ | ✓ |
| Conserved structure 3′ to M4 splice junction [43, 44] | ✓ | | ✓ | ✓ | ✓ | ✓ | | | ✓ | ✓ | ✓ | ✓ | ✓ | ✓ | ✓ | ✓ |
| M constrained region with vRNA structure ca. nt200—see main text [27] | ✓ | | | | ✓ | ✓ | ✓ | ✓ | ✓ | ✓ | ✓ | ✓ | ✓ | ✓ | ✓ | ✓ |
| M unknown constrained region ca. nt400 | | | | ✓ | | | ✓ | | | | | | | | ✓ | |
| M unknown constrained region ca. nt500 | | | | | | | | | | ✓ | | | | | | |
| Structured region 5′ to M2 splice acceptor [45–47] | | | | | | | ✓ | | | | | | ✓ | | ✓ | ✓ |
| M unknown constrained region ca. nt800; possible artefact | | | | | | | | | ✓ | | | | | | ✓ | |
| M 3′ packaging-associated [40, 41] | | | ✓ | | | | ✓ | ✓ | | | | | | | ✓ | |
| NS 5′ near packaging-associated [48, 49] | ✓ | ✓ | ✓ | ✓ | ✓ | | | ✓ | ✓ | | ✓ | | ✓ | ✓ | ✓ | ✓ |
| NS2 splice donor | ✓ | ✓ | ✓ | ✓ | ✓ | | ✓ | | ✓ | | ✓ | | ✓ | ✓ | ✓ | ✓ |
| NS1 region (ca. nt80–150) secondary structure [27, 43, 50, 51] | | | | | | ✓ | | | ✓ | | ✓ | ✓ | ✓ | ✓ | ✓ | ✓ |
| NS2 post-splice structure (possible splice acceptor structure) [52] | ✓ | ✓ | ✓ | ✓ | ✓ | ✓ | ✓ | ✓ | ✓ | ✓ | ✓ | ✓ | ✓ | ✓ | ✓ | ✓ |
| NS1/2 ORF overlap post-NS2 splice acceptor | ✓ | ✓ | ✓ | ✓ | ✓ | ✓ | ✓ | ✓ | | | ✓ | ✓ | ✓ | ✓ | ✓ | ✓ |
| Possible artefact following NS1 termination | ✓ | | | | | | ✓ | | | | | | | | | |

(*Continued*)

**Table 1.** (Continued)

| Description of constrained region | Human hosts | | | | | | Swine | | Avian hosts | | | | | | | |
|---|---|---|---|---|---|---|---|---|---|---|---|---|---|---|---|---|
| | H1N1 | | H3N2 | | H7N9 | | H1N2 | | H5N1 | | H5N8 | | H7N7 | | H7N9 | |
| | raw | rnk | raw | rnk | raw | rnk | raw | rnk | raw | rnk | raw | rnk | raw | rnk | raw | rnk |
| NS 3′ packaging-associated [48] | ✓ | | | ✓ | ✓ | | ✓ | ✓ | | ✓ | | | ✓ | ✓ | ✓ | |

Nucleotide (nt) positions, and 5′/3′ descriptions, refer to coding (+ sense) RNA. Columns marked "raw" correspond to analyses undertaken using unranked (raw) scores; columns marked "rnk" correspond to analyses undertaken using ranked scores (see Methods for details).

There is a strong consensus for codon 28 of PB2 encoding methionine, across all eight subtype/host combinations. This codon is the third methionine within PB2 (counting the canonical initiation site as the first), and has a stronger initiation context than the second. An alternative explanation for the constraint observed around codon 28, therefore, is that it is an alternative initiation site for an N-truncated form of PB2.

Bearing in mind that this is a packaging-associated region, we postulate that these predicted secondary structures have functions in packaging.

**A highly conserved stem-loop in PB2 5′ vRNA, adjacent to a set of alternative conserved motifs.**   Structural prediction using RNAalifold on the 5′ vRNA (corresponding to 3′ cRNA) regions deemed constrained in PB2 by RNAdescent predicts a well-conserved stem-loop (S1 Appendix, Fig B). The loop corresponds to codons 753 and 754 of PB2, equivalently NC_007373.1 nucleotides 2284–2289. Inspection of the per-codon output from RNAdescent shows that codons 751, 752, 754 and 755, all of which are predicted to form part of the stem, have high constraint across all subtypes, with the exception of codon 755 in the H1N1pdm09 sequences, where the lower constraint can be explained by a minority of sequences containing a codon that would be predicted to form a G-U base pair in the stem instead of a G-C base pair. Codon 753, which falls entirely within the loop (i.e. is not predicted to base pair locally), has a lower level of constraint, although again most of the reduced constraint can be explained by a G→A mutation in the wobble position rather than any other mutation that could encode an arginine. This may be explained by the generally low usage of CpG-containing codons to encode arginine, but also raises the possibility that the negative-sense loop has non-local base pairing, with some sequences forming a G-U instead of a G-C pair.

This region is independently predicted to be constrained in another analysis that combines SHAPE and DMS experiments with data on per-nucleotide variability [27], and a stem-loop with slightly different base pairing is predicted via another set of SHAPE experiments, in which it is also shown that inhibition of this region via locked nucleic acids impairs viral function *in vitro* and in a mouse model [26].

The region we predict to contain this stem loop has been investigated in three sets of mutational analysis experiments. In two sets of experiments [9, 25], synonymous mutations are made that dramatically reduce packaging. The synonymous mutations made are different in the two sets of experiments, but in both cases are mutations we would expect to disrupt our predicted stem-loop. In the third set of experiments [24], the relevant region is deleted entirely, again with a dramatic reduction in packaging resulting. We therefore postulate that the reason for the observed constraint in this location is the presence of a stem-loop critical to packaging, such that packaging is nearly abrogated when the stem-loop is disrupted or deleted.

Details of predicted structures are discussed further in S1 Appendix.

**A small conserved stem-loop in PB1 3′ vRNA.**   Only two RNAdescent analyses find constraint in the regions previously identified as packaging-associated in PB1 5′ cRNA

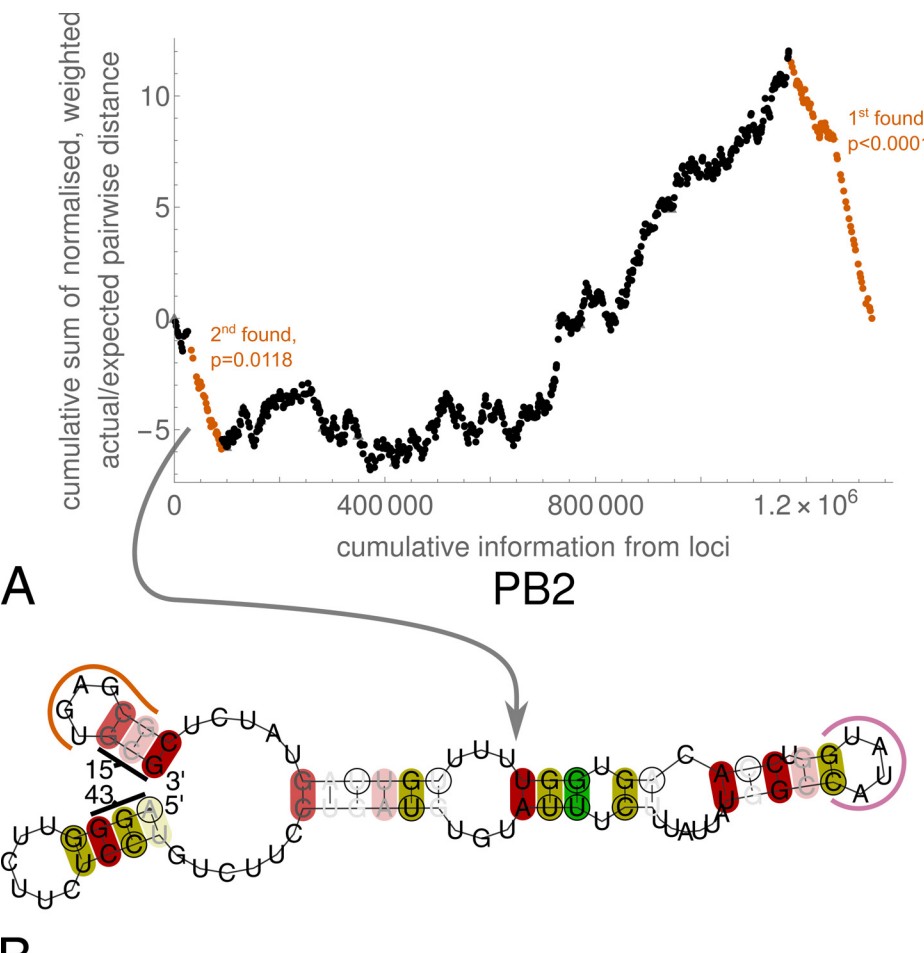

**Fig 1. PB2 3′ vRNA region whose cRNA is deemed constrained by RNAdescent. A** Example of graphical RNAdescent output, for analysis of raw constraint data in H5N8 PB2. Each dot/triangle represents a codon, with descents representing more constrained regions (see Methods and references [11, 12] for further details). Regions whose constraint is deemed significant by the algorithm are highlighted in orange, with manual annotation of the order in which the regions have been found and their bootstrap *p*-values. In this example, two constrained regions are found, corresponding to the packaging-associated regions at the ends of the vRNA. Plots for other combinations of subtype/host/type of data analysed may be found in S1 Code–S8 Code. **B** RNAalifold output from folding the PB2 3′ vRNA corresponding to the region of H5N8 cRNA deemed constrained in the RNAdescent analysis using raw constraint data. The fold is of the (− sense) vRNA, produced using RNAalifold [17, 18]. The termini of the predicted structure are labelled with the corresponding amino acid number of PB2. Portions of the two predicted stem-loops described in the main text are highlighted: an orange line highlights the nucleotides corresponding to codons 16 and 17 of PB2, and a purple line highlights the nucleotides corresponding to codons 27 and 28 of PB2. Base pairs are highlighted in deep/mid/light red when all/all but one/all but two sequences are capable of forming the pairs shown. Base pairs are highlighted in deep/mid/light yellow when all/all but one/all but two sequences are capable of forming the pair shown or one other pair (including GU pairs). Base pairs are highlighted in green when all sequences are capable of forming the pair shown or one of two other pairs (including GU pairs). RNAalifold was used with input options disallowing lonely pairs, allowing G-quadruplexes, and with the ribosum scoring matrix enabled. This avian strain fold was produced with the temperature set to 41°C. Fold predictions for other subtype/host/RNAdescent analysis type combinations may be found in Fig A of S1 Appendix.

(corresponding to 3′ vRNA), of which one is found only after accounting for other potentially interfering signals of constraint. A small stem-loop is identified in the RNAalifold predictions for both regions (Fig 2). Inspection of the per-codon RNAdescent output and of the individual codon usage in all strain/host combinations is consistent with the stem-loop forming in all

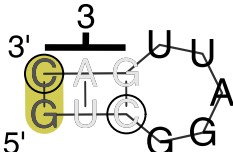   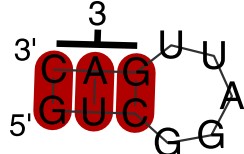

H5N1 avian (ranked)         H7N9 avian (ranked)

**Fig 2. Folds of PB1 3′ vRNA regions whose cRNA is deemed conserved by RNAdescent.** A small conserved stem-loop can be seen. Labels correspond to the influenza A subtype, host, and analysis method (raw constraint data or ranked constraint data) that predicted the constrained region. Folds are of the (− sense) vRNA, produced using RNAalifold. The folds displayed are truncated to the regions containing information on the stem-loop we discuss, using base pair predictions from the folds of the full regions predicted to be constrained; the full region folds may be found in S4 and S8 Figs. The folds are annotated with PB1 codon numbers. Base pairs are highlighted by RNAalifold in deep red when all sequences are capable of forming the pairs shown. Base pairs are highlighted in deep/mid yellow when all/all but one sequences are capable of forming the pair shown or one other pair (including GU pairs). RNAalifold was used with input options disallowing lonely pairs, allowing G-quadruplexes, and with the ribosum scoring matrix enabled. Avian strain folds were produced with the temperature set to 41˚C.

cases, raising the possibility that the conserved structure is present in all cases but because the feature covers only a small number of codons, it is difficult for the RNAdescent algorithm to find. A previous mutational analysis of this region [24] separately mutated three regions of RNA that cover the predicted stem-loop. In that study, packaging disruption was observed when these regions were mutated, and the extent of packaging disruption was ordered in a way that matches the extent of disruption that would be expected to our predicted structure.

A stem-loop exactly complementary to the stem-loop we predict in − sense is predicted in folds of the + sense RNA of the constrained regions, and the per-codon constraint is also consistent with the constraint that would be required for such a stem-loop. We note, however, that the requirement for − sense constraint is more consistent with the observed disruption to packaging when the region is mutated. We therefore postulate that our predicted−sense structure plays a role in packaging.

**A stem-loop in PB1 5′ vRNA partially, but not fully, explains high RNA constraint.**
RNAdescent analyses find constrained regions of varying length in the 3′ cRNA of PB1, almost all of which cover a region from codons 742–755 (NC_007372.1 nucleotides 2248–2289), and inspection of the RNAdescent output shows that in all cases, most individual codons within this region are particularly constrained. RNAalifold analyses of the constrained regions' vRNA predict a highly conserved stem-loop involving the nucleotides of codons 742–747 (S1 Appendix, Fig C). The individual codon constraint is higher in codons pairing within the stem in a manner consistent with formation of this stem-loop. The conserved stem-loop is independently predicted in another analysis that combines SHAPE and DMS experiments with data on per-nucleotide variability [27].

Two mutational analyses have previously been undertaken in this region. A mutational analysis by Marsh *et al.* [25], in which a substantial reduction in PB1 packaging was observed, would be expected to have disrupted the stem of the predicted stem-loop from codons 742–747, and so our proposed structure explains this previous result. A mutational analysis by Liang *et al.* [24] would be expected to have had minimal impact upon this predicted structure —this work and our prediction are not inconsistent, but neither are they mutually supportive.

**A consistently predicted stem-loop in PA 3′ vRNA but weak evidence of relationship to packaging.**   RNAalifold [17, 18] analyses of the constrained regions in PA 5′ cRNA identified by RNAdescent predict a conserved stem-loop in the regions corresponding to codons 9–14

(NC_007371.1 nucleotides 51–64; Fig 3). Individual codons in this region mostly have evidence of constraint in the RNAdescent analysis. The main exceptions are an AUG codon, where little information on RNA constraint is gained, and an apparently poorly-constrained codon 13 that usually encodes an isoleucine. Although codon 13 appears poorly constrained, inspection of the codon usage shows a bias towards using AUU or AUC in favour of AUA, and the two codons used both allow base pairing of the negative sense RNA with the U found on the predicted opposite strand of the stem, whereas the seldom-used codon would not: this exception to the RNA constraint observed in the region is therefore well explained if the predicted stem-loop is formed. Previous mutational analysis of this region (reference [24], mutations m10 and m11), introducing mutations that would be expected to disrupt the predicted stem-loop, has shown only a modest reduction in packaging efficiency in this region. However, in this previous mutational analysis, packaging efficiency was assayed by considering the ratio of GFP RNA from a PA-like construct to NA RNA, and so a mutation that affected packaging other than self-packaging of PA might not have been detected.

In coding sense, the region we have identified contains an in-frame AUG codon, which is the first AUG in frame after the canonical initiation site, and the second or third overall, depending upon sequence. Whilst an alternative explanation for constraint in the region with the predicted stem-loop is the presence of an alternative initiation site, there is not a strong Kozak context.

Previous evidence relating the 3′ region of PA vRNA (i.e. 5′ cRNA) to packaging [20, 24] establishes a relatively weak relationship between this region and packaging efficiency. Only the first few coding nucleotides have a strong association with packaging efficiency, raising the possibility that the non-coding region, which we have not investigated, has a major role to play. On balance, we do not have strong evidence to relate the structure we have predicted to packaging, but circumstantial evidence remains consistent with the possibility.

**Consistent identification of a region in PA 5′ vRNA not fully explained by previous packaging experiments.**   With a single exception (the H1N1 human host analysis with raw constraint data, which has the most challenging signal-to-noise ratios of all the datasets), our RNAdescent analyses consistently identify constrained regions in the 3′ cRNA of PA. Most of the constrained regions identified are around 100 nucleotides in length, and some are

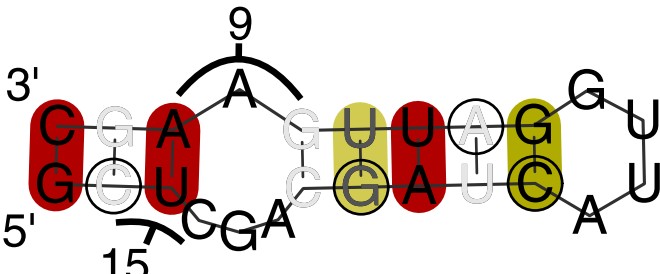

**Fig 3. Representative fold of stem-loop in PA 3′ vRNA region whose cRNA is deemed constrained by RNAdescent.** The fold displayed is of H7N9 avian host sequences, truncated to a region covering NC_026424.1 nucleotides 23–46, using base pair predictions from the folds of the full regions predicted to be constrained. The nucleotides corresponding to codons 9 and 15 are labelled. RNAalifold predicts similar folds for multiple analogous constrained regions in other subtype/host/RNAdescent analysis type combinations (S2, S6, S7 and S8 Figs), and one slightly differing fold in one analogue (S4 Fig). The displayed fold is of the−sense vRNA, produced using RNAalifold [17, 18]. Base pairs are highlighted by RNAalifold in deep red when all sequences are capable of forming the pairs shown. Base pairs are highlighted in deep/mid yellow when all/all but one sequences are capable of forming the pair shown or one other pair (including GU pairs). RNAalifold was used with input options disallowing lonely pairs, allowing G-quadruplexes, and with the ribosum scoring matrix enabled. Avian strain folds were produced with the temperature set to 41°C.

substantially longer. Our analysis output is strongly consistent with true constraint across this 100 nucleotide region, not artefact. The length of the identified constrained region contrasts with previous experiments on packaging [19], which established near-wildtype packaging efficiency when the 40 final coding nucleotides of PA were present alongside a sufficient length of the 5′ cRNA.

Presuming that constraint in the 40 coding nucleotides at the 3′ end of PA cRNA can be explained by a packaging requirement, this leaves a region of at least 60 nucleotides requiring explanation.

We begin by considering whether a packaging requirement could explain the additional constraint. The experiments undertaken by Liang *et al.* [19] that established the 40 nucleotide packaging-associated region determined packaging efficiency by comparing GFP expression from a truncated PA-GFP construct, with fluorescent mouse anti-NP antibody presence, in cells infected at low MOI. Later mutational analyses by Liang *et al.* [24] were constrained to the region previously determined to be necessary for packaging, and compared the quantity of GFP RNA in a mutated truncated PA-GFP construct with the quantity of NA RNA. A separate set of mutational analyses by Marsh *et al.* [25], which estimated packaging proportions of all segments by qPCR, found evidence that mutation in PA could impair packaging of other segments, including a substantial reduction in packaging efficiency of NP but a nil-to-modest reduction in packaging efficiency of NA. qPCR was performed after viral passaging, which may have selected for virions capable of making some protein products. In summary, previous experimental approaches do not provide direct evidence that a region longer than 40 nucleotides of the 3′ PA cRNA is required for efficient packaging, but do not exclude completely the possibility of a longer region being required for efficient packaging of other segments.

Were a packaging requirement not to explain the additional constraint, then an explanation for the additional constrained nucleotides would be required. In most cases, there is an open +1 frame in the region, terminating around nucleotide 2080. Around nucleotide 1970, there is a reasonably, but not entirely, conserved GGGGUUUU motif. Submotifs of this motif have previously been associated with frameshifts (reviewed in reference [53]), but most frameshifts associated with this motif are of the −1 type, and the position of termination codons in the −1 frame would not explain the overall constraint. Previous structural probes of the cRNA [21] suggest the possibility of coding nucleotides 2014–2037 forming a stem-loop; this stem-loop contains the motif CUU_AGG_G, which although not previously described, is similar to the CUU_AGG_C +1 slippery site motif proposed in southern tomato virus (STV) [54], and in fact would be more conducive to the anticodon 3′UCC remaining engaged after slippage than would be seen in the STV case.

Possible positions of negative-sense open reading frames are not entirely consistent with the observed constrained region, and Kozak contexts for possible initiation codons are weak. From a (− sense) vRNA structural perspective, RNAalifold analyses of the constrained regions in many cases predict conserved stem-loop motifs corresponding to PA codons 685–689 and 701–704 (see Fig 4), and individual codon constraint information from the RNAdescent analyses is consistent with such structures. However, given the evidence from mutational analysis experiments [25] that the region corresponding to PA codons 701–704 may be involved in the packaging of other segments, single-stranded structural predictions should be treated with caution. In general, structural predictions for the constrained regions do not give an overall sufficiently consistent prediction for the previously unrecognised region of constraint to draw conclusions as to why the entire region is found to be constrained (S1–S8 Figs).

In summary, the region we have found here as consistently constrained across multiple analyses is larger than can be explained by previous work, and hence determining an explanation for the constraint in this region is a target for future study.

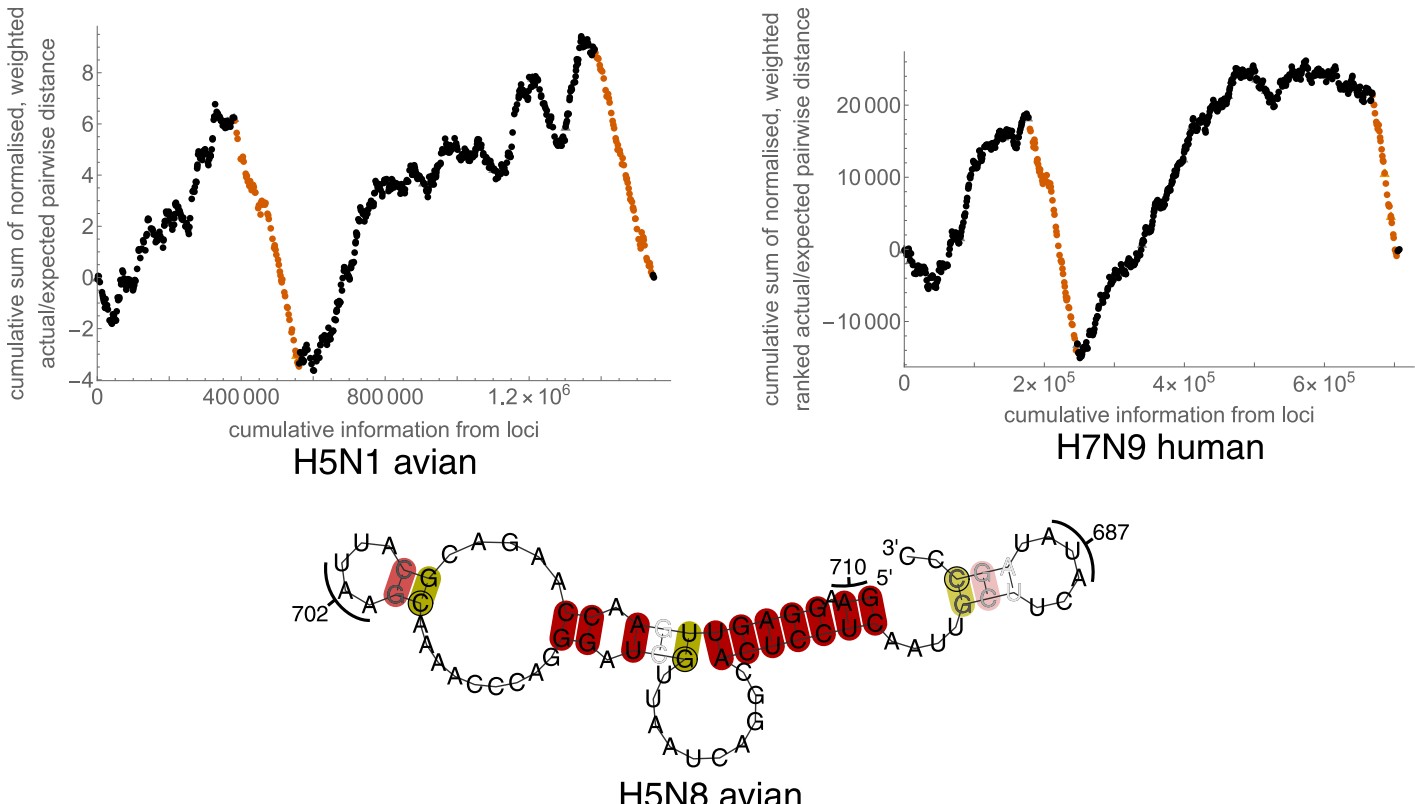

**Fig 4. Constraint analysis of PA 3′ cRNA (5′ vRNA) region.** Top: Examples of RNAdescent analysis output, for unweighted (raw) constraint data for H5N1 avian host sequences and for weighted constraint data for H7N9 human host sequences. Each point corresponds to one codon in PA, ordered 5′ to 3′ in cRNA. Codons that do not yield any information (weight zero) are plotted with open triangles. Orange points correspond to codons in regions deemed significantly constrained by the algorithm. In both cases, a steep descent can be seen in the centre of the plot corresponding to the conserved PA/PA-X out of frame overlap. A steep descent can also be seen to the 3′ side of the plot, indicating the long constrained regions found. Constraint is apparent throughout the entire regions. Bottom: Fold of (− sense) vRNA, from RNAalifold analysis of H5N8 avian host sequences. The region folded is the region of interest found in the 3′ cRNA from analysing ranked codon constraint data. To the right of the figure, a stem-loop corresponding to codons 685–689 may be seen. To the left of the figure, a stem-loop corresponding to codons 701–704 may be seen. The fold is of the −sense vRNA, produced using RNAalifold. Selected nucleotides are numbered with corresponding amino acid numbers. Base pairs are highlighted by RNAalifold in deep red when all sequences are capable of forming the pairs shown. Base pairs are highlighted in deep/mid yellow when all/all but one sequences are capable of forming the pair shown or one other pair (including GU pairs). RNAalifold was used with input options disallowing lonely pairs, allowing G-quadruplexes, and with the ribosum scoring matrix enabled. The fold was produced with the temperature set to 41˚C.

**Constrained regions in HA 5′ vRNA have good intra-type but poor inter-type constraint, raising a question regarding reassortment potential.**   Our RNAdescent analysis finds constrained regions in the 3′ cRNA (5′ vRNA) of the HA segments in all analysed subtype/host combinations (in all cases where ranked constraint data were analysed, and in most cases where raw/unranked data were analysed). The region has previously been determined to be packaging-associated [31–33]. Although, as has previously been seen with RNAdescent analyses of sequence alignments with different overall levels of constraint [12], the exact delineation of the constrained regions varies slightly between our analyses, there is a striking commonality between the regions deemed constrained in our analyses and a single region found in two previous analyses, containing the nucleotides encoding amino acids 543–547 and 544–548 respectively of the A/WSN/33(H1N1) virus [32, 33]. In the first of these analyses, it was found to result in a marked abrogation not only of HA packaging, but also of packaging of PB1 and, to a lesser extent, PA and NP. In the second analysis, HA packaging was abrogated and an interaction with a PB2 packaging-associated region was found. As a result, we were motivated

to examine the sequences relating to this sub-region more closely, finding that, in many cases, this sub-region contained some of the individual codons with greatest constraint across each analysis.

Surprisingly, despite the high within subtype/host constraint of sequence, and the consistent determination of constraint of this region, there is substantial deviation between analogous sequences from different haemagglutinin types (Table 2). Recalling that our algorithm will detect RNA constraint that occurs separately from amino acid constraint, we considered the possibility that we have a region of relative RNA constraint on a background of pressure to diversify at amino acid level. However, our predicted constrained region encodes part of the transmembrane domain, meaning this is not a region of the protein usually considered to be of antigenic importance and under diversification pressure at amino acid level.

RNAalifold [17, 18] predicts that these regions are structured (S1 Appendix, Fig D), but there is not a single secondary structure predicted that is convincingly conserved between subtypes. A similar pattern of predicted structure that is inconsistent between subtypes is seen in the cRNA (S1–S8 Figs), and although structured cRNA has been predicted in this region previously [21], we find no particular between-subtype pattern to suggest that cRNA structural requirements particularly drive constraint.

Bearing in mind that previous experimental data implicate this region not only in self-packaging of HA, but also in other segments' packaging, we were motivated to examine whether the inconsistency of our structural predictions could be explained by their failing to take into account heterodimer (or possibly multimer) formation with other segments. We also considered the possibility that the lack of consistent structural prediction points to structural reconfigurations in a region that is performing a chaperone function—and the possibility that a

**Table 2. Sequences of comparable constrained regions in the 5′ vRNA (3′ cRNA) packaging-associated region of HA.**

| Type | Host | nt ref | | aa ref | Sequence |
|---|---|---|---|---|---|
| H1N1 | human | NC_026433.1 | 1630 | 544 | gua guc ucc cUG Gg---g gca auc a---gc - |
| H1N2 | swine | AB731585.1 | 1630 | 544 | uua guc ucc cug gg---g gca auc a---gc - |
| H3N2 | human | NC_007366.1 | 1629 | 534 | --- auu ucc -uuu gcc au---A UCa ugu uuu |
| H5N1 | avian | NC_007362.1 | 1642 | 540 | -- agu -ucc cua gca cug gca auc aug aug- |
| H5N8 | avian | LC699656.1 | 1646 | 540 | -- agu -ucc cua gca cug GCA AUc aug gug- |
| H7N7 | avian | KM922677.1 | 1609 | 539 | --- cuu --- cug gcc auU GCA --- aug ggc- |
| H7N9 | human | NC_026425.1 | 1609 | 537 | --- cuu --- cua gcc auU GUA --- aug ggc- |
| H7N9 | avian | NC_026425.1 | 1609 | 537 | --- cuu --- cua gcc auU GUA --- aug ggc- |

The consensus sequence (in + sense) is shown for each type/host combination. Where <90% of sequences in the collection have the consensus sequence nucleotide, the nucleotide is represented in a smaller font size. Codons falling within regions identified as highly constrained by RNAdescent on analysis of raw (unranked) constraint data are coloured blue; codons falling within regions identified as highly constrained on analysis of ranked constraint data are coloured orange; codons falling within regions identified as highly constrained on both analyses are coloured purple. Individual codons are underlined with a straight line if their normalised weighted raw score on RNAdescent analysis is < 0 (i.e. their constraint is above the weighted mean). Individual codons are underlined with a wavy line if their normalised weighted ranked score on RNAdescent analysis is < 0 (i.e. their constraint is above the weighted median). Where small stem-loops are predicted in vRNA (− sense) by RNAalifold analysis, nucleotides predicted to form part of the loops are capitalised. Nucleotide references ("nt ref") are those in the noted sequence of the first non-gap nucleotide represented. Codon references ("aa ref") are those in the consensus sequence (i.e. not counting any gaps in alignments used for analysis) for the type/host combination of the first non-gap codon. A manual alignment of the consensus sequences is displayed. Although the sequences are broadly comparable, the constraint of primary structure between HA types is surprisingly low for a region associated with packaging.

chaperone function could be performed by cRNA, as well as vRNA, if the cRNA were appropriately co-located with the molecules to be packaged. There are possible complements between this region of HA and regions we have identified in both PB2 and PB1. We discuss the potential interaction with PB2 in S1 Appendix, and consider the potential interaction with PB1 in the next paragraph.

With regard to potential interactions between constrained regions in HA and PB1, we note that the constrained regions in HA of the H3 and H5 strains contain (in + sense) UCAU motifs, and the H7 strains contain CCAU motifs, whose reverse complements would (allowing G-U pairs) base-pair with the constrained UCAU motif identified and predicted to form a loop in the PB1 5′ vRNA (S1 Appendix, Fig C). A similar compatible motif does not appear to be present in the HAs of H1 strains, but these strains (as well as the H5 strains) contain GAUC motifs whose (palindromic) reverse complements could base-pair with one of the palindromic GAUC motifs predicted to form the stem of the same PB1 stem-loop motifs.

The relatively low between-subtype conservation in the constrained HA regions, together with the existing experimental evidence and further computational evidence we present for between segment interactions involving this region, raises a key question of how efficiently HA segments could reassort between viruses. Naïvely, the between-segment base-pairings we have identified between HA and PB1 are sufficient to permit reassortment, although in practice it is unlikely that four critical bases only from each segment would be involved in dimerisation, and the involvement of additional RNA and/or nucleoprotein would be expected.

**A conserved stem-loop in NP 5′ vRNA.**  Constrained regions are identified by RNAdescent in the 3′ cRNA (corresponding to 5′ vRNA) of NP in almost all subtypes. Inspection of the RNAdescent output shows that almost all individual codons between 468 and 481 inclusive are highly constrained. Many RNAalifold predictions show a stem-loop in the vRNA (S1 Appendix, Fig E).

The predicted stem-loop is consistent with one predicted in another analysis that combines SHAPE and DMS experiments with data on per-nucleotide variability [27]. This region has previously been also predicted to contain a pseudoknot in the vRNA and a different stem-loop in the cRNA [21, 35, 37]. We note that the structure prediction algorithm we have used does not predict pseudoknots. Mutational analysis has previously shown this region to be important in packaging [36], but the mutations with most impact on packaging in the previous study bookend the region where we predict a stem-loop, and mutations of our region of interest were not attempted. Bearing in mind the effects on packaging of mutating nearby loci, we propose that our predicted stem-loop may have packaging functionality. Any investigation of this function would need to take into account the potential effect of mutational analysis experiments on other previously predicted structures.

**A highly conserved stem-loop in M 3′ vRNA.**  The first 200 coding nucleotides of the M segment contain at least two initiation sites, two splice sites, and associated secondary structure, as well as being a region associated with packaging. It is, therefore, unsurprising that this region or subregions within it are deemed constrained in all RNAdescent analyses (all subtype/host/constraint measure combinations). Whether constrained regions are identified separately or contiguously depends upon the relative signal/noise in each analysis. However, the separate/contiguous identification affects the size of the regions our pipeline presents to the folding algorithm, and hence the length scale that implicitly dominates when folds are predicted.

Shorter folds of the (− sense) vRNA in the region corresponding to M1 codons 17–23 predict a stem-loop (S1 Appendix, Fig F). Although fold predictions of longer regions do not predict this stem-loop, in all analyses the raw output from RNAdescent shows that these codons constitute a highly-constrained subregion, supportive of the stem-loop prediction.

Codons 18–20 of M1 have previously been experimentally synonymously mutated, with a marked reduction in viral packaging efficiency and infectivity [40]. The nucleotides corresponding to codons 18 and 19 are predicted to fall on one side of the stem, and the mutations made would be expected to be disruptive of the predicted stem. The nucleotides corresponding to codon 20 are predicted to form the loop. Although their constraint would not be required to maintain the predicted structure or amino acid usage (leucine), they are nonetheless highly constrained.

We therefore postulate that this region forms a stem-loop and that the structure is required for packaging. The structural prediction could be probed by attempting rescue of the previously observed disruption via compensatory mutation on the opposite strand of the predicted stem (noting that compensatory mutation would generate a premature termination codon and so disrupt protein production). Following verification of the structural prediction, mutation of codon 20 in isolation would probe with which regions the structure interacts.

**A consistently predicted M vRNA stem-loop corresponding to M1 codons 65–72 may be associated with packaging.**   RNAdescent analyses covering every subtype/host combination, and all but two of our analyses overall, deem the region corresponding to M1 codons 65–72 to lie within regions with significant constraint. In every such case, RNAalifold analysis of secondary structure predicts a conserved stem-loop in the (– sense) vRNA (Fig 5). In all subtype/host combinations, raw analysis output from RNAdescent shows high constraint in each of codons 65–72, with a small number of individual exceptions that can all be explained by mutations that would not disrupt pairing (mutations that would cause G-U → A-U pairing, mutations that would cause G-C → G-U pairing, or mutations of a wobble position not predicted to form part of the stem-loop)—that is, the mutations that are observed are supportive of base-pairing consistent with the predicted structure. In a number of cases, codons 73–77 are also highly constrained, which cannot be explained by this structural prediction.

A similarly structured region has previously been predicted by a study with a different approach to structural determination [27].

A previous set of experiments by Ozawa *et al.* [41], using M segments with central deletions and an in-frame GFP insertion, demonstrated an approximately four-fold increase in infectious virus-like particles when 222, compared with 111, nucleotides of the M 5′ cRNA were included. The packaging efficiency, as determined by GFP expression, was comparable.

The consistent packaging *efficiency* of M pseudo-segments when this region is included/excluded, notwithstanding the change in *infectiousness*, may point to a function of this region

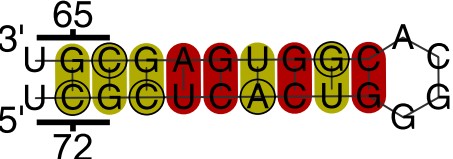

**Fig 5. Predicted stem-loop in M vRNA in the region corresponding to M1 codons 65–72.** The image shows a truncation to the region of interest (nucleotides corresponding to codons 65 and 72 marked) of the RNAalifold prediction of the constrained region arising from RNAdescent analysis of H5N8 avian host sequences, using raw (unranked) constraint data. All RNAdescent analyses, except those of H1N1 human host sequences with ranked constraint data and H3N2 human host sequences with raw constraint data, predict constraint in this region and the base pairings displayed here are replicated in all other RNAalifold predictions. (Some RNAalifold analyses predict one or two additional base pairings that elongate the stem.) Base pairs are highlighted by RNAalifold in deep red when all sequences are capable of forming the pairs shown. Base pairs are highlighted in deep yellow when all sequences are capable of forming the pair shown or one other pair (including GU pairs). RNAalifold was used with input options disallowing lonely pairs, allowing G-quadruplexes, and with the ribosum scoring matrix enabled. Avian strain folds were produced with the temperature set to 41˚C.

other than M self-packaging. However, the consistent prediction of a structure in the vRNA, together with an increase in infectiousness, points towards a function of the RNA in this region. The structure being in the vRNA points towards a packaging function, and a function in packaging of non-self segments has not been investigated.

## Constrained regions associated with alternative protein expression

Many of the constrained regions found by RNAdescent that are not associated with packaging can be associated with splicing, frameshifting, or expression of truncated proteins. One region that is not discussed here is discussed in S1 Appendix.

**PB1-N92 initiation region in avian strains.**   A series of alternative initiation sites in the PB1 segment has previously been investigated [55, 56]. In addition to the PB1-F2 overlapping open reading frame, these sites include short ORFs in-frame with PB1-F2, as well as a version of PB1 with a 40aa N-terminal truncation ("PB1-N40").

Previous investigations have noted the strong Kozak context around, but have not extended to investigating, the protein that would be generated by a 92aa N-terminal truncation of PB1 ("PB1-N92") [55]. Bearing in mind the practicalities of a ribosome reading as far as such an initiation site [57], expression of such a truncated protein seems *a priori* less plausible than expression of the canonical protein or of alternatives with smaller truncations. Our algorithm finds constrained regions covering or near this Kozak sequence, in precisely the virus strains investigated from avian hosts (see Table 1 for summary).

It is important to recognise that whether a constrained region is found in a particular viral strain depends in part upon the number of viruses sampled and the genetic diversity between individual samples (this explains why, in general, fewer constrained regions are found in H1N1pdm09 viruses from humans, which diverged relatively recently). It is therefore not possible from our results to draw a firm conclusion that there is something particular to avian influenza A strains that requires a greater level of constraint in this region than is required in mammalian strains. However, the pattern of constraint raises this possibility, and motivates further investigation of how the Kozak contexts vary between strains and whether secondary RNA structure may play a role.

Comparison of the Kozak contexts for the strains (Fig 6) shows that whilst there are characteristic differences between strains at some nucleotide positions, all strains have a strong Kozak context. We have produced predicted folds for all of the investigated strains (S1 Appendix, Figs G and H): there is not a clear pattern that dichotomises between the avian and the mammalian strains. Finally, noting the predicted presence of a pseudoknot in the initiation regions for PB1-F2 and PB1-N40 [58], we used the RNAPKplex pseudoknot predictor [18] on the consensus sequences for each strain covering the regions folded during the RNAalifold investigation. No pseudoknots were predicted.

Overall, we conclude that although there is constraint in these regions, and the constraint is likely to be realised in RNA structure, it is unclear from our results alone to what extent these factors relate to initiation of a truncated PB1-N92 protein. In conjunction with further work on PB1-N92 initiation, our results may, however, point towards further avenues of investigation.

**Candidate PA-X frameshift stimulator.**   The +1 frameshift during PA translation to give the PA-X C-terminal alternative protein is now well-established [10, 61]. Much of the discussion regarding the mechanism of frameshifting has focussed on the presence of a slowly-decoded CGU codon in the ribosomal A-site stimulating slippage in the P-site. Our constraint analysis suggests an alternative/additional possibility, driven by the structure of the RNA itself.

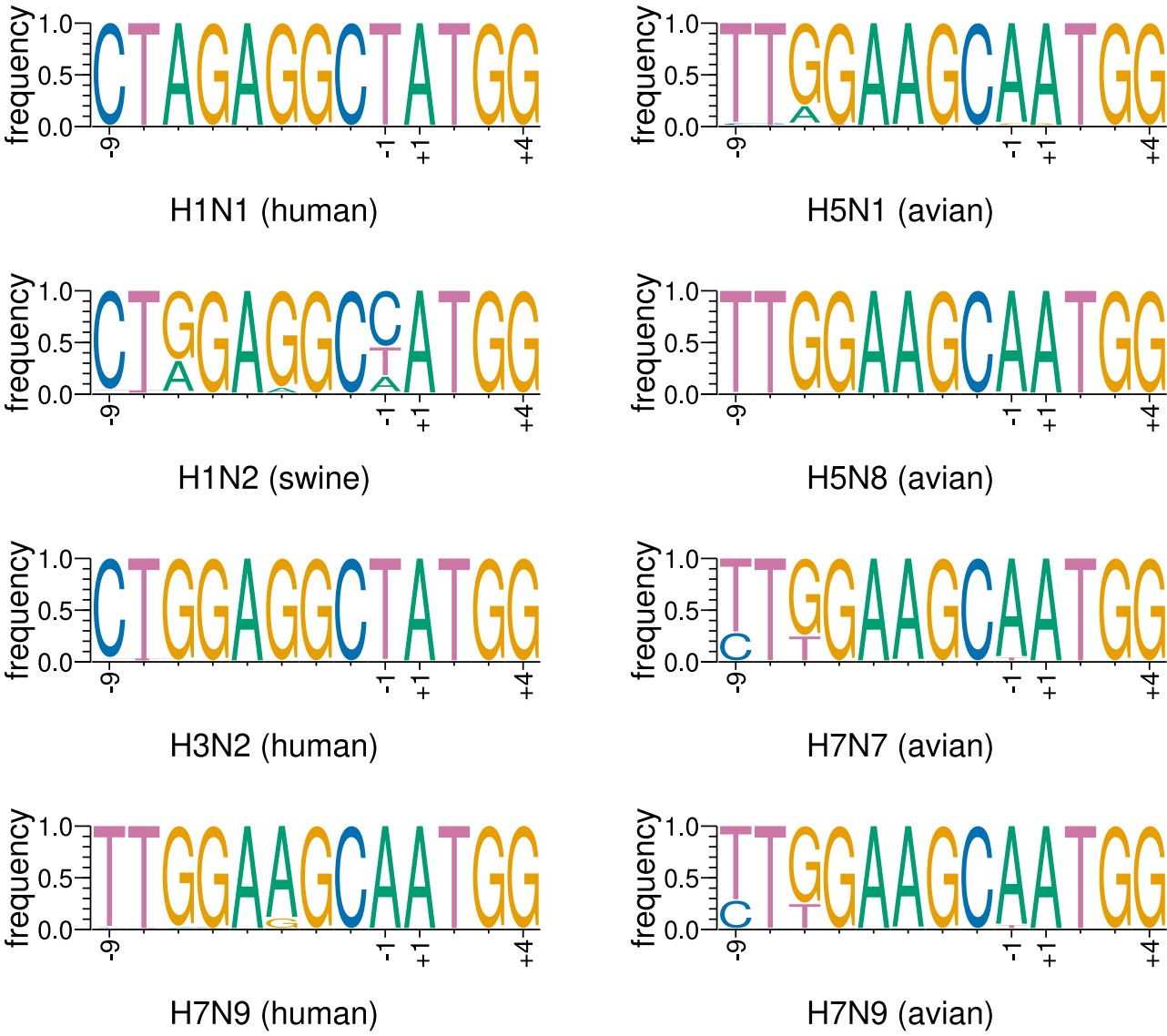

**Fig 6. Kozak contexts of PB1-N92 initiation codons in investigated mammalian and avian strains.** Although characteristic differences can be seen between individual codons for mammalian versus avian strains, all strains have strong initiation contexts. Note that the H7N9 (human) viruses have more in common with avian than with other mammalian viruses, as might be expected for a strain where infection in humans is predominantly a direct zoonosis. Plots were generated using WebLogo3 [59, 60].

Whilst many of the constrained regions output by our RNAdescent analysis of the PA/PA-X out-of-frame overlap contain the entire overlap, a substantial number contain the frameshift motif and a short region 3′ to it. Modified minimum free energy folds of these regions consistently predict stem-loops, with the frameshift motif toward the base of the 5′ stem (Fig 7). The size of the stem-loop predicted varies, depending upon which of the 3′ GAAA motifs pairs with UUUC from the frameshift motif; in some predictions, the shortest stem-loop has a GC pair, which may increase the stability of the stem-loop [62]. Although most +1 ribosomal frameshifting is postulated to occur via tRNA slippage, in which ribosomal slippage occurs whilst the ribosome is paused by a codon with rare tRNA usage [63], −1 frameshifting involving pausing attributable to stem-loop motifs is well-described [64]. We propose that at

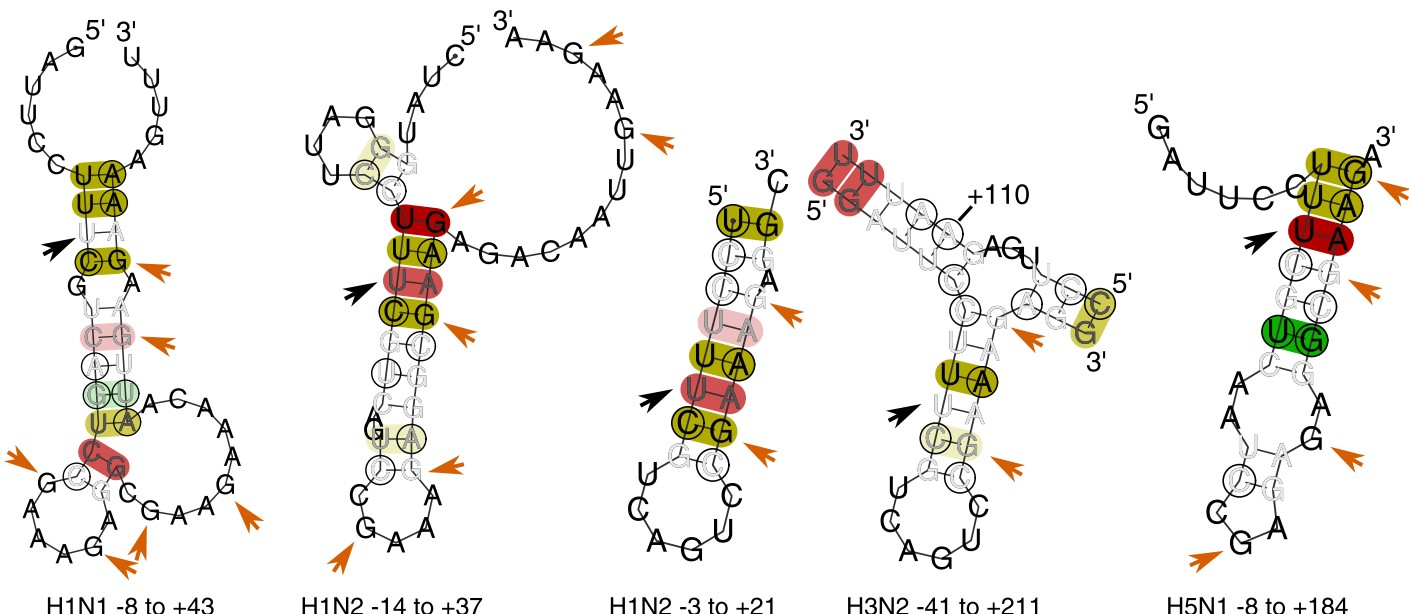

**Fig 7. Examples of predicted structures near the PA/PA-X frameshift.** Predictions are shown using RNAalifold [18] on sections of the alignments for the labelled subtypes, with the fourth position in the slippery site UCC_UU**U**_CGU marked with black arrows and numbered "+1". Folding was performed on subsequences beginning before and ending after this position as indicated in the labels; these subsequences correspond to regions deemed constrained by our algorithm. Where a long region was folded, only the portion of the fold containing the slippery site and stem-loop is displayed (where this results in discontinuous display of a portion of nucleic acid, a nucleotide is labelled to give its position in relation to the slippery site). Possible GAAA (in some cases with an A substituted by G) motifs that can base pair with the UUUC of the slippery site are marked in orange. A number of possible base pairings leading to stem-loop motifs are predicted. We postulate that an ensemble of such stem-loops is in fact seen, with the composition of the ensemble capable of modifying the relative PA/PA-X abundances (see main text): the differing predicted secondary structures would therefore arise because small differences in the input data result in the algorithm used reaching different, but numerically very close, modified free energy minima. Predicted structures for analysed but not shown subtypes have similar topology to one of the structures displayed (S3–S8 Figs).

least two (and possibly more) stem-loop structures form, involving the PA slippery site and a region 3′ to it, and leading to ribosomal pausing. The effect of ribosomal pausing would be expected to vary depending upon conformation—with a shorter stem-loop, pausing would occur with the frameshift motif in the A-site, whereas with a longer stem-loop, the frameshift motif would have left the ribosome. In this way, conformational switching in the RNA would lead to a greater tendency to produce PA or PA-X. Such an effect could be additive with pausing from the described usage of the rare CGU codon in this region [10, 61].

We therefore propose that the entire region up to 45 nucleotides 3′ of the PA-X frameshift site functions as a "frameshift stimulator", adopting different conformations that are more or less likely to lead to ribosomal frameshifting.

Conserved structure in this region has been suggested previously [21], although not in ensemble form and not in association with frameshift regulation.

**Splicing-associated regions.** Three splice donors and two (possibly three) splice acceptors have been described across the M and NS segments, with associated RNA structures [42, 45, 46, 52, 65–67]. These regions are found in many of our constraint analyses.

**Region in the NS intron important for NS1 expression.** A region in NS just downstream of the NS2 splice donor has previously been identified as structured, and mutation of the cRNA associated with reduced expression of NS1 protein [43, 50, 51], indicating importance of this region in splicing. At least part of this region is identified as constrained in RNAdescent analyses covering five of our eight subtype/host combinations. However, RNAalifold predictions for the structures of the constrained regions do not contain structures that are clearly

**Fig 8. Possible stem-loop structure in the NS intron.** Left: The displayed sequence is the H5N8 consensus sequence, with putative base pairs from manual inspection. NS1 codons 28–30 and 35–38 have high levels of constraint. The UCCU motif in the loop is complementary to the NS2 splice acceptor site; we note that some sequences contain an alternative UUCU motif (annotated in blue), which is still capable of pairing. Right: RNAalifold prediction of pairing in H7N7 sequences, using as input the region predicted by analysis of ranked constraint data. Base pairs are highlighted in deep/mid red when all/all but one sequences are capable of forming the pairs shown. Base pairs are highlighted in deep yellow when all sequences are capable of forming the pair shown or own other pair (including GU pairs). RNAalifold was used with input options disallowing lonely pairs, allowing G-quadruplexes, and with the ribosum scoring matrix enabled. The H7N7 fold was produced with the temperature set to 41˚C. Other structures predicted by RNAalifold may be found in S4–S8 Figs.

constrained between the subtype/host combinations, and the structural predictions do not fully agree with those of earlier publications, all of which involved pre-2009 H1N1 strains.

We performed inspection of our detailed codon-level constraint information to gain additional insight into which of the predicted structures correlates well with our constraint output. In most subtype/host combinations, high constraint is seen at nucleotide level in codons 28–30 and 35–38. The consensus sequence in this region can fold into a stem-loop with a very high number of G-C pairs (Fig 8). Many of the non-consensus sequences have point mutations that would still allow base pairing (either predicted to be unpaired, or with a canonical/non-canonical pairing switch), although there are some sequences with entirely different primary structures in this region. A structure with these pairings is observed in the RNAalifold prediction for ranked analysis of H7N7 sequences, with two additional pairings within the loop—we emphasise that although this secondary structure is consistent with the primary structures seen in other subtype/host combinations, RNAalifold predicts alternate secondary structures for those combinations. We note that if this stem-loop were formed, the larger loop (without in-loop pairings) would contain a sequence complementary to the splice acceptor site, and this may be capable of inhibiting spliceosome activity as the loop is brought into the acceptor site's proximity along with the donor site. Such inhibition would be particularly useful if the stem-loop structure formed under some, but not all, circumstances, so that it could function as a switch directing relative abundances of splicing products. This would be consistent with a minimum free energy approach, such as that of RNAalifold, predicting multiple structures when given slightly different inputs.

A previous mutational analysis experiment [50] deleted part of this region, in a way that would have disrupted the putative stem-loop structure, and if the structure had been inhibiting splicing, then a relative reduction in NS1 species, as was seen, might be expected. The earlier experiment did not measure abundance of NS2 species, to determine whether the NS1 results could be explained by a change in relative mRNA abundance. It would be possible to investigate this potential role in splicing via point mutation/rescue experiments, bearing in mind that the richness of G-C pairing means that individual mutations may not completely disrupt the predicted structure.

## Structured regions lacking functional explanations, including previously undescribed regions

Our analysis reproduces some structured regions in influenza A RNA currently without a full functional explanation that have previously been described. We also find constraint in some

regions that to our knowledge have not been described in the literature. We discuss one of these regions in more detail below, and the remainder in S1 Appendix.

**Stem-loop motifs in haemagglutinin associated with transition to high-pathogenicity avian influenza.**   In just one of our analyses (ranked score analysis of H7N9 human host viruses), a constrained region is identified near nt1000 in the HA segment. This region has previously been identified as containing conserved stem-loops [28–30], with the possibility of polymerase slippage leading to chain elongation near the HA cleavage site, and the development of accessory cleavage sites that extend cellular tropism and may be associated with high-pathogenicity avian influenza strains. Fig 9 shows our structural prediction for this region.

We note that whilst our algorithm is set up not to penalise insertions (so as to allow motifs that may or may not be present within a constrained region), the existence of itself of a motif with a tendency towards the occurrence of insertions should not lead to a region being called as constrained. It is likely that the algorithm is picking up the constraint in the region that leads to the known stem-loop motifs. However, the underlying explanation for this constraint remains unknown. The region deemed constrained by the algorithm is near the threshold for identification, meaning random variation could suffice to explain why the region is deemed constrained in one subtype associated with transition to high pathogenicity and not deemed constrained in others.

## Discussion

This extensive *in silico* analysis of a wealth of experimentally generated influenza A sequence data has, in a number of places, allowed us to provide structural explanations for previous experimental observations, and to propose further investigations that will corroborate these explanations. In other regions of the genome, where previous understanding is less advanced, we have highlighted areas suitable for further investigation. Our work continues a years-long synergystic exchange in influenza A research between mathematical/computational analysis and bench investigation, in which the analysis has made possible bench investigations that would have been infeasible without additional insight, and results from bench investigations have guided interpretation of analytical results. Many of these previous results additionally constitute useful positive controls for our findings. We anticipate that our analysis will enable the design of novel experimental approaches to investigate the molecular biology of this important human and veterinary pathogen.

In some of the regions where we have described constraint (e.g. regions associated with splicing/frameshifting), confirmation of our functional predictions would immediately lead to candidate targets for functional inhibition of the influenza lifecycle and hence a pathway for novel antiviral discovery.

In packaging-associated regions, the need to characterise interactions further is likely to make the investigational pathway to an antiviral longer. Nonetheless, given how critical packaging is to the viral lifecycle, the ability to inhibit packaging will remain a valuable goal worth pursuing. It is likely that, as we begin to improve our characterisation of interactions required for packaging, we shall gain insight into reassortment potential of influenza A viruses and hence their pandemic potential. In this work, we have moved from delineating regions involved in packaging towards explaining the structural basis behind these regions, and so we have enabled further steps towards characterising reassortment and pandemic potential. We would caution that although we have suggested structural bases for particular regions of RNA being involved in packaging, we have not fully characterised the interactions of structures we predict to be important. Such interactions will likely involve the nucleoprotein, and may involve other RNA elements, both of which may themselves influence the structures formed.

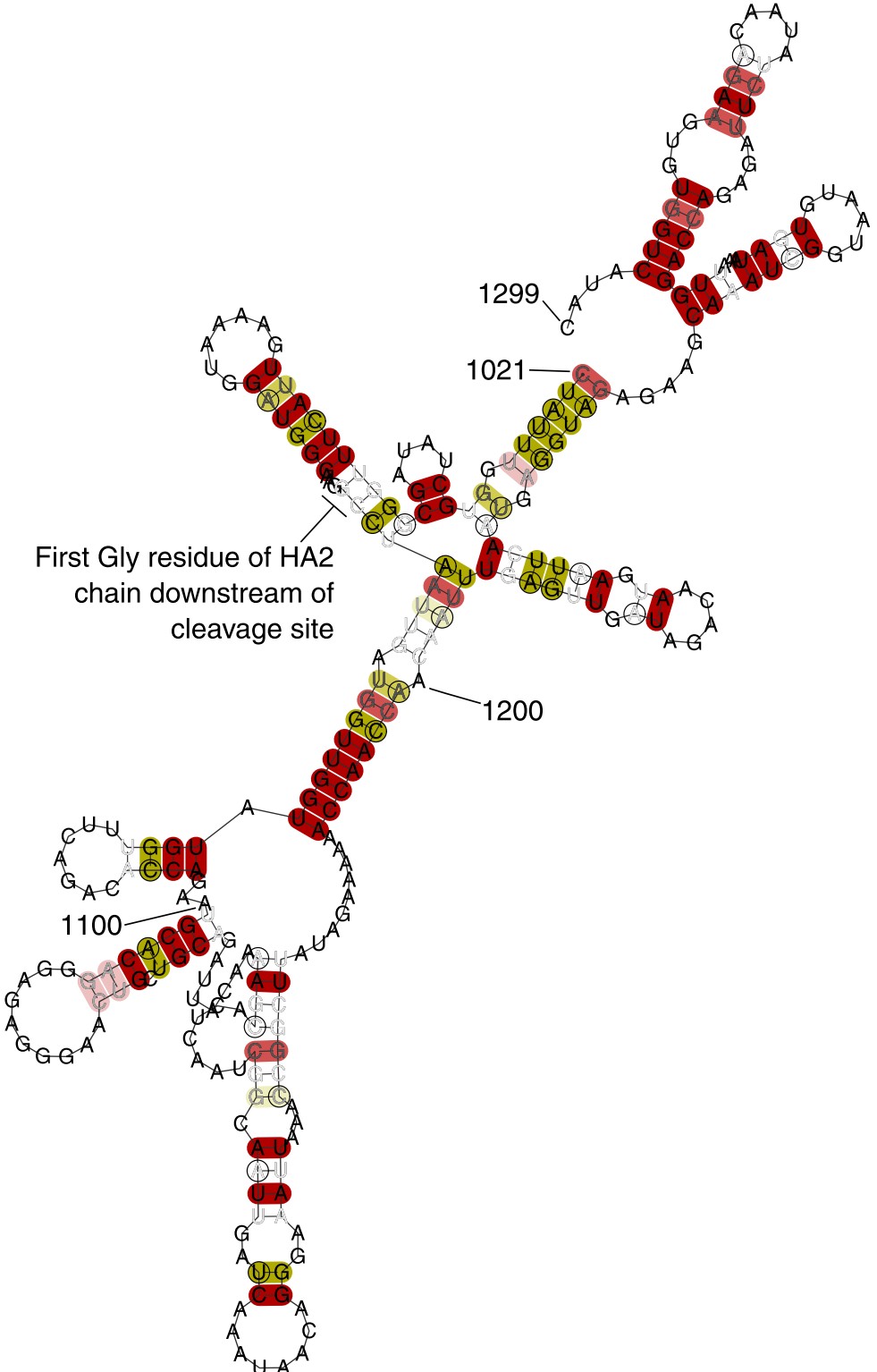

**Fig 9. RNAalifold structure prediction of the conserved stem-loops near the HA cleavage site in H7N9 human host viruses.** Nucleotide numbering follows RefSeq NC_026425.1 (GenBank KF021597.1). RNAalifold [18] uses a minimum free energy algorithm, modified to take into account how many sequences in an alignment can form predicted base pairs. Red (light red, pink)/yellow (light yellow, pale yellow) highlighted base pairs indicate that all (all but one, all but two) sequences in the alignment can form the base pair, with no/one alternative base pairing. We

reproduce the small previously predicted [28] stem-loop starting at nt1030; although we predict a larger stem-loop near the cleavage site, it differs from stem-loops predicted for other strains.

Full computational prediction of nucleic acid–nucleoprotein interactions is yet to be solved, and should be regarded as a limitation of our methodology.

Our work has not specifically addressed vaccine design. We nonetheless note the challenge in developing vaccine strains of ensuring that a vaccine retains sufficient immunogenicity whilst, for a live vaccine, attenuating pathogenicity. Our work highlighting regions requiring high constraint in wild-type viruses may therefore be useful when investigating the performance of vaccine strains, as comparison of sequences may generate explanations for attenuation.

Our work has highlighted a plethora of candidate regions for experimental investigation, and we acknowledge the need for such investigations to confirm our findings.

Sufficient influenza A sequence data are now available to conduct separate analyses by subtype and host, and to make comparisons between subtype/host combinations. By analysing viruses from different subtypes and hosts, we have been able to corroborate findings between different datasets. However, this approach has also highlighted some limitations in our methodology. Although our approach is designed to find constrained regions in a scale-agnostic fashion, the designation of a region as "constrained" or "not constrained" does not fully reflect the subtlety that within a constrained region, some constraint may be more important than other constraint. Our inspection of codon-level results to some extent addresses this issue, but further methodological development that takes into account more highly-constrained subregions would give greater granularity to results. This is a particularly pertinent issue for secondary structural prediction, where the inclusion of two nearby constrained subregions in a single algorithmically determined region may result in an inappropriate length scale being selected to prime a folding algorithm.

The use of ranked constraint data, rather than the raw data, in constraint analyses is aimed at minimising artefact in data whose test statistics do not approach the central (Gaussian) limit closely [12]. In our study of influenza A data, we have examples of where the use of ranking can find additional signal through the minimisation of artefact, and where the use of ranking through reduction in granularity misses signal that can be seen from raw data, *within data from the same dataset* (e.g. the terminal packaging-associated signals in H5N1 NS data). That is, which analysis method can be "best" at finding a signal of constrained nucleotides can depend sufficiently on local variation patterns rather than overall dataset variability that running just one of the analyses may miss constrained regions within a single dataset. Rather than relying upon graphical observations or formal testing to decide on the more appropriate test to run, it may therefore be more appropriate in future to run both types of analysis on each dataset, and use such observations/testing *post hoc* to help resolve discordant results.

A key limitation of our work is that we have analysed coding regions. Given the potential importance of UTRs in viral lifecycles, switching to a metric capable of analysing constraint in non-coding regions will be important in future. However, beyond the methodological issue of comparing nucleic acid constraint between coding and non-coding regions is a more pertinent experimental issue, of obtaining reliable UTR sequencing data in quantity. Analysing UTRs therefore represents a different problem in data acquisition and processing, likely to exceed any challenge in developing further analytical methodology.

Some decisions regarding which data to include in each of our analyses are constrained by the metadata available in the underlying database. Although we have aimed to use sequences

representative of natural infections in the pertinent hosts, in a number of cases viruses have been passaged to amplify the genetic material available prior to sequencing, and excluding all passaged viruses would remove a substantial amount of data, particularly from earlier isolates. We have therefore retained some passaged isolates. Information about passaging in the underlying database is not always recorded consistently, meaning it is not possible to be certain that all passaged isolates we might wish to exclude have indeed been excluded. However, the database that we have used contains more usable metadata on passaging than some large public databases, allowing a greater level of curation.

Because we have subdivided analyses only by recorded haemagglutinin, neuraminidase, and host types, it is possible that in our analyses of other viral segments we have included reassorted segments. The consensus method for classifying influenza A viruses uses only haemagglutinin and neuraminidase sequences, so grouping other segments other than by their associated HA/NA sequences would involve a unilateral development or selection of a classification scheme. Our final manual curation checking step is likely to have removed some such reassortants, leaving only reassortants whose sequences align well with other sequences in a given dataset. This means that any remaining reassortant sequences should not have unduly disrupted the algorithmic steps of our analysis. From a biological perspective, regardless of whether the genetic variability in a sequence arises from point mutations or from reassortment, the sequence still enables evaluation of the evolutionary constraints on a virus capable of infecting a given host.

Some limitations in our structural predictions are inherited from the abilities and limitations of the software we have chosen to make these predictions. For example, the software only reports a single modified minimum free energy structure (that is, the structure with the lowest value of predicted free energy plus an adjustment term that takes into account the number of sequences in an alignment capable of forming the predicted structure), meaning it is difficult to evaluate whether multiple structures are likely to exist in ensemble. Clues to this possibility may be seen where different structural predictions are made for different subtype/host analyses, as described for possible structures near the PA/PA-X frameshift. We emphasise that for cases where we have displayed multiple predicted structures for different subtypes, there is the possibility that small differences in data have caused artefactual differences in the minimum free energy predictions from the algorithm, in addition to the possibilities that these represent different selections from an ensemble of truly occurring structures, and that there is true difference between the structures adopted by different subtypes. Conversely, however, this gives us greater confidence in predicted structures that are robust to perturbation of the exact input data: our presentation of different predicted structures should be viewed primarily as an opportunity to identify highly stable elements likely to be of significance in the viral lifecycle. A different example of a software limitation is that the software only predicts structures using multiple sequence alignments for a single strand, making it more difficult to evaluate the feasibility of long-range interactions. Multiple possible long-range interactions have previously been described [68, 69]; we note that although we are limited in evaluating these via structural prediction algorithms, none of the long-range interactions predicted across different wild-type viruses consists of a pairing of regions of RNA where RNAdescent predicts both regions forming the pair to be constrained. This may be unsurprising, given that the authors of the previous works found that mutation of viruses led to viable viruses exhibiting different long-range interactions, meaning that constraint was not crucial for viability. Overall, although we have chosen secondary structure prediction software missing some features that in some contexts may be helpful, we believe that it is logical to follow methodology that uses the properties of multiple sequence alignments to predict regions of constraint with structural predictions using the entire alignments. Software that can make such predictions and that has other potentially desirable features is not currently available.

## Supporting information

**S1 Appendix. Additional notes on regions found to have nucleotide-level constraint.**
(PDF)

**S1 Table. Number of sequences used in evaluations for each gene/subtype/host combination.** Note that H1N1 sequences contain a non-functional PB1-F2 and so no PB1-F2 analysis was performed for these sequences.
(PDF)

**S2 Table. Summary of regions of significant constraint found in H1N1pdm09 (human host) influenza A genes, using weighted and raw (unranked) codon variability values.** Reference sequences used are RefSeq NC_026438.1 (GenBank FJ984387.1), NC_026435.1 (GQ323558.1), NC_026437.1 (FJ969539.1), NC_026433.1 (FJ969540.1), NC_026436.1 (FJ969536.1), NC_026434.1 (FJ984386.1), NC_026431.1 (FJ969537.1), NC_026432.1 (FJ969538.1), for segments 1–8, respectively. Citation details may be found in S1 Appendix. *Denotes a region only found by excluding a potentially interfering signal. *Z*- and *p*-values in parentheses denote values prior to removal of the next most significant signal. If parenthetical values are absent, then such a signal was removed in an earlier step only.
(PDF)

**S3 Table. Summary of regions of significant constraint found in H1N2 (swine host) influenza A genes, using weighted and raw (unranked) codon variability values.** Reference sequences used are GenBank AB731582.1, AB731583.1, AB731584.1, AB731585.1, AB731586.1, AB731587.1, AB731588.1, AB731589.1, for segments 1–8, respectively. Citation details may be found in S1 Appendix. *Denotes a region only found by excluding a potentially interfering signal. *Z*- and *p*-values in parentheses denote values prior to removal of the next most significant signal. If parenthetical values are absent, then such a signal was removed in an earlier step only.
(PDF)

**S4 Table. Summary of regions of significant constraint found in H3N2 (human host) influenza A genes, using weighted and raw (unranked) codon variability values.** Reference sequences used are RefSeq NC_007373.1 (GenBank CY002071.1), NC_007372.1 (CY002070.1), NC_007371.1 (CY002069.1), NC_007366.1 (CY002064.1), NC_007369.1 (CY002067.1), NC_007368.1 (CY002066.1), NC_007367.1 (CY002065.1), NC_007370.1 (CY002068.1), for segments 1–8, respectively. Citation details may be found in S1 Appendix. *Denotes a region only found by excluding a potentially interfering signal. *Z*- and *p*-values in parentheses denote values prior to removal of the next most significant signal. If parenthetical values are absent, then such a signal was removed in an earlier step only.
(PDF)

**S5 Table. Summary of regions of significant constraint found in H5N1 (avian host) influenza A genes, using weighted and raw (unranked) codon variability values.** Reference sequences used are RefSeq NC_007357.1 (GenBank AF144300.1), NC_007358.1 (AF144301.1), NC_007359.1 (AF144302.1), NC_007362.1 (AF144305.1), NC_007360.1 (AF144303.1), NC_007361.1 (AF144304.1), NC_007363.1 (AF144306.1), NC_007364.1 (AF144307.1), for segments 1–8, respectively. Citation details may be found in S1 Appendix. *Denotes a region only found by excluding a potentially interfering signal. *Z*- and *p*-values in parentheses denote values prior to removal of the next most significant signal. If parenthetical values are absent, then such a signal was removed in an earlier step only.
(PDF)

**S6 Table. Summary of regions of significant constraint found in H5N8 (avian host) influenza A genes, using weighted and raw (unranked) codon variability values.** Reference sequences used are GenBank LC699653.1, LC699654.1, LC699655.1, LC699656.1, LC699657.1, LC699658.1, LC699659.1, LC699660.1, for segments 1–8, respectively. Citation details may be found in S1 Appendix. *Denotes a region only found by excluding a potentially interfering signal. *Z*- and *p*-values in parentheses denote values prior to removal of the next most significant signal. If parenthetical values are absent, then such a signal was removed in an earlier step only.
(PDF)

**S7 Table. Summary of regions of significant constraint found in H7N7 (avian host) influenza A genes, using weighted and raw (unranked) codon variability values.** Reference sequences used are GenBank KM922674.1, KM922675.1, KM922676.1, KM922677.1, KM922678.1, KM922679.1, KM922680.1, KM922673.1, for segments 1–8, respectively. Citation details may be found in S1 Appendix. *Denotes a region only found by excluding a potentially interfering signal. *Z*- and *p*-values in parentheses denote values prior to removal of the next most significant signal. If parenthetical values are absent, then such a signal was removed in an earlier step only.
(PDF)

**S8 Table. Summary of regions of significant constraint found in H7N9 (human host) influenza A genes, using weighted and raw (unranked) codon variability values.** Reference sequences used are RefSeq NC_026422.1 (GenBank KF021594.1), NC_026423.1 (KF021595.1), NC_026424.1 (KF021596.1), NC_026425.1 (KF021597.1), NC_026426.1 (KF021598.1), NC_026429.1 (KF021599.1), NC_026427.1 (KF021600.1), NC_026428.1 (KF021601.1), for segments 1–8, respectively. Citation details may be found in S1 Appendix. *Denotes a region only found by excluding a potentially interfering signal. *Z*- and *p*-values in parentheses denote values prior to removal of the next most significant signal. If parenthetical values are absent, then such a signal was removed in an earlier step only.
(PDF)

**S9 Table. Summary of regions of significant constraint found in H7N9 (avian host) influenza A genes, using weighted and raw (unranked) codon variability values.** Reference sequences used are RefSeq NC_026422.1 (GenBank KF021594.1), NC_026423.1 (KF021595.1), NC_026424.1 (KF021596.1), NC_026425.1 (KF021597.1), NC_026426.1 (KF021598.1), NC_026429.1 (KF021599.1), NC_026427.1 (KF021600.1), NC_026428.1 (KF021601.1), for segments 1–8, respectively. Citation details may be found in S1 Appendix. *Denotes a region only found by excluding a potentially interfering signal. *Z*- and *p*-values in parentheses denote values prior to removal of the next most significant signal. If parenthetical values are absent, then such a signal was removed in an earlier step only.
(PDF)

**S10 Table. Summary of regions of significant constraint found in H1N1pdm09 (human host) influenza A genes, using weighted and ranked codon variability values.** Reference sequences used are RefSeq NC_026438.1 (GenBank FJ984387.1), NC_026435.1 (GQ323558.1), NC_026437.1 (FJ969539.1), NC_026433.1 (FJ969540.1), NC_026436.1 (FJ969536.1), NC_026434.1 (FJ984386.1), NC_026431.1 (FJ969537.1), NC_026432.1 (FJ969538.1), for segments 1–8, respectively. Citation details may be found in S1 Appendix. *Denotes a region only found by excluding a potentially interfering signal. *Z*- and *p*-values in parentheses denote values prior to removal of the next most significant signal. If parenthetical values are absent, then

such a signal was removed in an earlier step only.
(PDF)

**S11 Table. Summary of regions of significant constraint found in H1N2 (swine host) influenza A genes, using weighted and ranked codon variability values.** Reference sequences used are GenBank AB731582.1, AB731583.1, AB731584.1, AB731585.1, AB731586.1, AB731587.1, AB731588.1, AB731589.1, for segments 1–8, respectively. Citation details may be found in S1 Appendix. *Denotes a region only found by excluding a potentially interfering signal. *Z*- and *p*-values in parentheses denote values prior to removal of the next most significant signal. If parenthetical values are absent, then such a signal was removed in an earlier step only.
(PDF)

**S12 Table. Summary of regions of significant constraint found in H3N2 (human host) influenza A genes, using weighted and ranked codon variability values.** Reference sequences used are RefSeq NC_007373.1 (GenBank CY002071.1), NC_007372.1 (CY002070.1), NC_007371.1 (CY002069.1), NC_007366.1 (CY002064.1), NC_007369.1 (CY002067.1), NC_007368.1 (CY002066.1), NC_007367.1 (CY002065.1), NC_007370.1 (CY002068.1), for segments 1–8, respectively. Citation details may be found in S1 Appendix. *Denotes a region only found by excluding a potentially interfering signal. *Z*- and *p*-values in parentheses denote values prior to removal of the next most significant signal. If parenthetical values are absent, then such a signal was removed in an earlier step only.
(PDF)

**S13 Table. Summary of regions of significant constraint found in H5N1 (avian host) influenza A genes, using weighted and ranked codon variability values.** Reference sequences used are RefSeq NC_007357.1 (GenBank AF144300.1), NC_007358.1 (AF144301.1), NC_007359.1 (AF144302.1), NC_007362.1 (AF144305.1), NC_007360.1 (AF144303.1), NC_007361.1 (AF144304.1), NC_007363.1 (AF144306.1), NC_007364.1 (AF144307.1), for segments 1–8, respectively. Citation details may be found in S1 Appendix. *Denotes a region only found by excluding a potentially interfering signal. *Z*- and *p*-values in parentheses denote values prior to removal of the next most significant signal. If parenthetical values are absent, then such a signal was removed in an earlier step only.
(PDF)

**S14 Table. Summary of regions of significant constraint found in H5N8 (avian host) influenza A genes, using weighted and ranked codon variability values.** Reference sequences used are GenBank LC699653.1, LC699654.1, LC699655.1, LC699656.1, LC699657.1, LC699658.1, LC699659.1, LC699660.1, for segments 1–8, respectively. Citation details may be found in S1 Appendix. *Denotes a region only found by excluding a potentially interfering signal. *Z*- and *p*-values in parentheses denote values prior to removal of the next most significant signal. If parenthetical values are absent, then such a signal was removed in an earlier step only.
(PDF)

**S15 Table. Summary of regions of significant constraint found in H7N7 (avian host) influenza A genes, using weighted and ranked codon variability values.** Reference sequences used are GenBank KM922674.1, KM922675.1, KM922676.1, KM922677.1, KM922678.1, KM922679.1, KM922680.1, KM922673.1, for segments 1–8, respectively. Citation details may be found in S1 Appendix. *Denotes a region only found by excluding a potentially interfering signal. *Z*- and *p*-values in parentheses denote values prior to removal of the next most

significant signal. If parenthetical values are absent, then such a signal was removed in an earlier step only.
(PDF)

**S16 Table. Summary of regions of significant constraint found in H7N9 (human host) influenza A genes, using weighted and ranked codon variability values.** Reference sequences used are RefSeq NC_026422.1 (GenBank KF021594.1), NC_026423.1 (KF021595.1), NC_026424.1 (KF021596.1), NC_026425.1 (KF021597.1), NC_026426.1 (KF021598.1), NC_026429.1 (KF021599.1), NC_026427.1 (KF021600.1), NC_026428.1 (KF021601.1), for segments 1–8, respectively. Citation details may be found in S1 Appendix. [*]Denotes a region only found by excluding a potentially interfering signal. *Z*- and *p*-values in parentheses denote values prior to removal of the next most significant signal. If parenthetical values are absent, then such a signal was removed in an earlier step only.
(PDF)

**S17 Table. Summary of regions of significant constraint found in H7N9 (avian host) influenza A genes, using weighted and ranked codon variability values.** Reference sequences used are RefSeq NC_026422.1 (GenBank KF021594.1), NC_026423.1 (KF021595.1), NC_026424.1 (KF021596.1), NC_026425.1 (KF021597.1), NC_026426.1 (KF021598.1), NC_026429.1 (KF021599.1), NC_026427.1 (KF021600.1), NC_026428.1 (KF021601.1), for segments 1–8, respectively. Citation details may be found in S1 Appendix. [*]Denotes a region only found by excluding a potentially interfering signal. *Z*- and *p*-values in parentheses denote values prior to removal of the next most significant signal. If parenthetical values are absent, then such a signal was removed in an earlier step only.
(PDF)

**S18 Table. GISAID acknowledgement table, containing list of sequences used, for RNAdescent analysis of PB2 sequences from H1N1pdm09 viruses originating from human hosts.**
(XLSX)

**S19 Table. GISAID acknowledgement table, containing list of sequences used, for RNAdescent analysis of PB1 sequences from H1N1pdm09 viruses originating from human hosts.**
(XLSX)

**S20 Table. GISAID acknowledgement table, containing list of sequences used, for RNAdescent analysis of PA sequences from H1N1pdm09 viruses originating from human hosts.**
(XLSX)

**S21 Table. GISAID acknowledgement table, containing list of sequences used, for RNAdescent analysis of PA-X sequences from H1N1pdm09 viruses originating from human hosts.**
(XLSX)

**S22 Table. GISAID acknowledgement table, containing list of sequences used, for RNAdescent analysis of HA sequences from H1N1pdm09 viruses originating from human hosts.**
(XLSX)

**S23 Table. GISAID acknowledgement table, containing list of sequences used, for RNAdescent analysis of NP sequences from H1N1pdm09 viruses originating from human hosts.**
(XLSX)

**S24 Table. GISAID acknowledgement table, containing list of sequences used, for RNAdescent analysis of NA sequences from H1N1pdm09 viruses originating from human hosts.**
(XLSX)

**S25 Table. GISAID acknowledgement table, containing list of sequences used, for RNAdescent analysis of M1 sequences from H1N1pdm09 viruses originating from human hosts.**
(XLSX)

**S26 Table. GISAID acknowledgement table, containing list of sequences used, for RNAdescent analysis of M2 sequences from H1N1pdm09 viruses originating from human hosts.**
(XLSX)

**S27 Table. GISAID acknowledgement table, containing list of sequences used, for RNAdescent analysis of NS1 sequences from H1N1pdm09 viruses originating from human hosts.**
(XLSX)

**S28 Table. GISAID acknowledgement table, containing list of sequences used, for RNAdescent analysis of NS2 sequences from H1N1pdm09 viruses originating from human hosts.**
(XLSX)

**S29 Table. GISAID acknowledgement table, containing list of sequences used, for RNAdescent analysis of PB2 sequences from H1N2 viruses originating from swine hosts.**
(XLSX)

**S30 Table. GISAID acknowledgement table, containing list of sequences used, for RNAdescent analysis of PB1 sequences from H1N2 viruses originating from swine hosts.**
(XLSX)

**S31 Table. GISAID acknowledgement table, containing list of sequences used, for RNAdescent analysis of PB1-F2 sequences from H1N2 viruses originating from swine hosts.**
(XLSX)

**S32 Table. GISAID acknowledgement table, containing list of sequences used, for RNAdescent analysis of PA sequences from H1N2 viruses originating from swine hosts.**
(XLSX)

**S33 Table. GISAID acknowledgement table, containing list of sequences used, for RNAdescent analysis of PA-X sequences from H1N2 viruses originating from swine hosts.**
(XLSX)

**S34 Table. GISAID acknowledgement table, containing list of sequences used, for RNAdescent analysis of HA sequences from H1N2 viruses originating from swine hosts.**
(XLSX)

**S35 Table. GISAID acknowledgement table, containing list of sequences used, for RNAdescent analysis of NP sequences from H1N2 viruses originating from swine hosts.**
(XLSX)

**S36 Table. GISAID acknowledgement table, containing list of sequences used, for RNAdescent analysis of NA sequences from H1N2 viruses originating from swine hosts.**
(XLSX)

**S37 Table. GISAID acknowledgement table, containing list of sequences used, for RNAdescent analysis of M1 sequences from H1N2 viruses originating from swine hosts.**
(XLSX)

**S38 Table. GISAID acknowledgement table, containing list of sequences used, for RNAdescent analysis of M2 sequences from H1N2 viruses originating from swine hosts.**
(XLSX)

**S39 Table. GISAID acknowledgement table, containing list of sequences used, for RNAdescent analysis of NS1 sequences from H1N2 viruses originating from swine hosts.**
(XLSX)

**S40 Table. GISAID acknowledgement table, containing list of sequences used, for RNAdescent analysis of NS2 sequences from H1N2 viruses originating from swine hosts.**
(XLSX)

**S41 Table. GISAID acknowledgement table, containing list of sequences used, for RNAdescent analysis of PB2 sequences from H3N2 viruses originating from human hosts.**
(XLSX)

**S42 Table. GISAID acknowledgement table, containing list of sequences used, for RNAdescent analysis of PB1 sequences from H3N2 viruses originating from human hosts.**
(XLSX)

**S43 Table. GISAID acknowledgement table, containing list of sequences used, for RNAdescent analysis of PB1-F2 sequences from H3N2 viruses originating from human hosts.**
(XLSX)

**S44 Table. GISAID acknowledgement table, containing list of sequences used, for RNAdescent analysis of PA sequences from H3N2 viruses originating from human hosts.**
(XLSX)

**S45 Table. GISAID acknowledgement table, containing list of sequences used, for RNAdescent analysis of PA-X sequences from H3N2 viruses originating from human hosts.**
(XLSX)

**S46 Table. GISAID acknowledgement table, containing list of sequences used, for RNAdescent analysis of HA sequences from H3N2 viruses originating from human hosts.**
(XLSX)

**S47 Table. GISAID acknowledgement table, containing list of sequences used, for RNAdescent analysis of NP sequences from H3N2 viruses originating from human hosts.**
(XLSX)

**S48 Table. GISAID acknowledgement table, containing list of sequences used, for RNAdescent analysis of NA sequences from H3N2 viruses originating from human hosts.**
(XLSX)

**S49 Table. GISAID acknowledgement table, containing list of sequences used, for RNAdescent analysis of M1 sequences from H3N2 viruses originating from human hosts.**
(XLSX)

**S50 Table. GISAID acknowledgement table, containing list of sequences used, for RNAdescent analysis of M2 sequences from H3N2 viruses originating from human hosts.**
(XLSX)

**S51 Table. GISAID acknowledgement table, containing list of sequences used, for RNAdescent analysis of NS1 sequences from H3N2 viruses originating from human hosts.**
(XLSX)

**S52 Table. GISAID acknowledgement table, containing list of sequences used, for RNAdescent analysis of NS2 sequences from H3N2 viruses originating from human hosts.**
(XLSX)

**S53 Table. GISAID acknowledgement table, containing list of sequences used, for RNAdescent analysis of PB2 sequences from H5N1 viruses originating from avian hosts.**
(XLSX)

**S54 Table. GISAID acknowledgement table, containing list of sequences used, for RNAdescent analysis of PB1 sequences from H5N1 viruses originating from avian hosts.**
(XLSX)

**S55 Table. GISAID acknowledgement table, containing list of sequences used, for RNAdescent analysis of PB1-F2 sequences from H5N1 viruses originating from avian hosts.**
(XLSX)

**S56 Table. GISAID acknowledgement table, containing list of sequences used, for RNAdescent analysis of PA sequences from H5N1 viruses originating from avian hosts.**
(XLSX)

**S57 Table. GISAID acknowledgement table, containing list of sequences used, for RNAdescent analysis of PA-X sequences from H5N1 viruses originating from avian hosts.**
(XLSX)

**S58 Table. GISAID acknowledgement table, containing list of sequences used, for RNAdescent analysis of HA sequences from H5N1 viruses originating from avian hosts.**
(XLSX)

**S59 Table. GISAID acknowledgement table, containing list of sequences used, for RNAdescent analysis of NP sequences from H5N1 viruses originating from avian hosts.**
(XLSX)

**S60 Table. GISAID acknowledgement table, containing list of sequences used, for RNAdescent analysis of NA sequences from H5N1 viruses originating from avian hosts.**
(XLSX)

**S61 Table. GISAID acknowledgement table, containing list of sequences used, for RNAdescent analysis of M1 sequences from H5N1 viruses originating from avian hosts.**
(XLSX)

**S62 Table. GISAID acknowledgement table, containing list of sequences used, for RNAdescent analysis of M2 sequences from H5N1 viruses originating from avian hosts.**
(XLSX)

**S63 Table. GISAID acknowledgement table, containing list of sequences used, for RNAdescent analysis of NS1 sequences from H5N1 viruses originating from avian hosts.**
(XLSX)

**S64 Table. GISAID acknowledgement table, containing list of sequences used, for RNAdescent analysis of NS2 sequences from H5N1 viruses originating from avian hosts.**
(XLSX)

**S65 Table. GISAID acknowledgement table, containing list of sequences used, for RNAdescent analysis of PB2 sequences from H5N8 viruses originating from avian hosts.**
(XLSX)

**S66 Table. GISAID acknowledgement table, containing list of sequences used, for RNAdescent analysis of PB1 sequences from H5N8 viruses originating from avian hosts.**
(XLSX)

**S67 Table. GISAID acknowledgement table, containing list of sequences used, for RNAdescent analysis of PB1-F2 sequences from H5N8 viruses originating from avian hosts.**
(XLSX)

**S68 Table. GISAID acknowledgement table, containing list of sequences used, for RNAdescent analysis of PA sequences from H5N8 viruses originating from avian hosts.**
(XLSX)

**S69 Table. GISAID acknowledgement table, containing list of sequences used, for RNAdescent analysis of PA-X sequences from H5N8 viruses originating from avian hosts.**
(XLSX)

**S70 Table. GISAID acknowledgement table, containing list of sequences used, for RNAdescent analysis of HA sequences from H5N8 viruses originating from avian hosts.**
(XLSX)

**S71 Table. GISAID acknowledgement table, containing list of sequences used, for RNAdescent analysis of NP sequences from H5N8 viruses originating from avian hosts.**
(XLSX)

**S72 Table. GISAID acknowledgement table, containing list of sequences used, for RNAdescent analysis of NA sequences from H5N8 viruses originating from avian hosts.**
(XLSX)

**S73 Table. GISAID acknowledgement table, containing list of sequences used, for RNAdescent analysis of M1 sequences from H5N8 viruses originating from avian hosts.**
(XLSX)

**S74 Table. GISAID acknowledgement table, containing list of sequences used, for RNAdescent analysis of M2 sequences from H5N8 viruses originating from avian hosts.**
(XLSX)

**S75 Table. GISAID acknowledgement table, containing list of sequences used, for RNAdescent analysis of NS1 sequences from H5N8 viruses originating from avian hosts.**
(XLSX)

**S76 Table. GISAID acknowledgement table, containing list of sequences used, for RNAdescent analysis of NS2 sequences from H5N8 viruses originating from avian hosts.**
(XLSX)

**S77 Table. GISAID acknowledgement table, containing list of sequences used, for RNAdescent analysis of PB2 sequences from H7N7 viruses originating from avian hosts.**
(XLSX)

**S78 Table. GISAID acknowledgement table, containing list of sequences used, for RNAdescent analysis of PB1 sequences from H7N7 viruses originating from avian hosts.**
(XLSX)

**S79 Table. GISAID acknowledgement table, containing list of sequences used, for RNAdescent analysis of PB1-F2 sequences from H7N7 viruses originating from avian hosts.**
(XLSX)

**S80 Table. GISAID acknowledgement table, containing list of sequences used, for RNAdescent analysis of PA sequences from H7N7 viruses originating from avian hosts.**
(XLSX)

**S81 Table. GISAID acknowledgement table, containing list of sequences used, for RNAdescent analysis of PA-X sequences from H7N7 viruses originating from avian hosts.**
(XLSX)

**S82 Table. GISAID acknowledgement table, containing list of sequences used, for RNAdescent analysis of HA sequences from H7N7 viruses originating from avian hosts.**
(XLSX)

**S83 Table. GISAID acknowledgement table, containing list of sequences used, for RNAdescent analysis of NP sequences from H7N7 viruses originating from avian hosts.**
(XLSX)

**S84 Table. GISAID acknowledgement table, containing list of sequences used, for RNAdescent analysis of NA sequences from H7N7 viruses originating from avian hosts.**
(XLSX)

**S85 Table. GISAID acknowledgement table, containing list of sequences used, for RNAdescent analysis of M1 sequences from H7N7 viruses originating from avian hosts.**
(XLSX)

**S86 Table. GISAID acknowledgement table, containing list of sequences used, for RNAdescent analysis of M2 sequences from H7N7 viruses originating from avian hosts.**
(XLSX)

**S87 Table. GISAID acknowledgement table, containing list of sequences used, for RNAdescent analysis of NS1 sequences from H7N7 viruses originating from avian hosts.**
(XLSX)

**S88 Table. GISAID acknowledgement table, containing list of sequences used, for RNAdescent analysis of NS2 sequences from H7N7 viruses originating from avian hosts.**
(XLSX)

**S89 Table. GISAID acknowledgement table, containing list of sequences used, for RNAdescent analysis of PB2 sequences from H7N9 viruses originating from human hosts.**
(XLSX)

**S90 Table. GISAID acknowledgement table, containing list of sequences used, for RNAdescent analysis of PB1 sequences from H7N9 viruses originating from human hosts.**
(XLSX)

**S91 Table. GISAID acknowledgement table, containing list of sequences used, for RNAdescent analysis of PB1-F2 sequences from H7N9 viruses originating from human hosts.**
(XLSX)

**S92 Table. GISAID acknowledgement table, containing list of sequences used, for RNAdescent analysis of PA sequences from H7N9 viruses originating from human hosts.**
(XLSX)

**S93 Table. GISAID acknowledgement table, containing list of sequences used, for RNAdescent analysis of PA-X sequences from H7N9 viruses originating from human hosts.**
(XLSX)

**S94 Table. GISAID acknowledgement table, containing list of sequences used, for RNAdescent analysis of HA sequences from H7N9 viruses originating from human hosts.**
(XLSX)

**S95 Table. GISAID acknowledgement table, containing list of sequences used, for RNAdescent analysis of NP sequences from H7N9 viruses originating from human hosts.**
(XLSX)

**S96 Table. GISAID acknowledgement table, containing list of sequences used, for RNAdescent analysis of NA sequences from H7N9 viruses originating from human hosts.**
(XLSX)

**S97 Table. GISAID acknowledgement table, containing list of sequences used, for RNAdescent analysis of M1 sequences from H7N9 viruses originating from human hosts.**
(XLSX)

**S98 Table. GISAID acknowledgement table, containing list of sequences used, for RNAdescent analysis of M2 sequences from H7N9 viruses originating from human hosts.**
(XLSX)

**S99 Table. GISAID acknowledgement table, containing list of sequences used, for RNAdescent analysis of NS1 sequences from H7N9 viruses originating from human hosts.**
(XLSX)

**S100 Table. GISAID acknowledgement table, containing list of sequences used, for RNAdescent analysis of NS2 sequences from H7N9 viruses originating from human hosts.**
(XLSX)

**S101 Table. GISAID acknowledgement table, containing list of sequences used, for RNAdescent analysis of PB2 sequences from H7N9 viruses originating from avian hosts.**
(XLSX)

**S102 Table. GISAID acknowledgement table, containing list of sequences used, for RNAdescent analysis of PB1 sequences from H7N9 viruses originating from avian hosts.**
(XLSX)

**S103 Table. GISAID acknowledgement table, containing list of sequences used, for RNAdescent analysis of PB1-F2 sequences from H7N9 viruses originating from avian hosts.**
(XLSX)

**S104 Table. GISAID acknowledgement table, containing list of sequences used, for RNAdescent analysis of PA sequences from H7N9 viruses originating from avian hosts.**
(XLSX)

**S105 Table. GISAID acknowledgement table, containing list of sequences used, for RNAdescent analysis of PA-X sequences from H7N9 viruses originating from avian hosts.**
(XLSX)

**S106 Table. GISAID acknowledgement table, containing list of sequences used, for RNAdescent analysis of HA sequences from H7N9 viruses originating from avian hosts.**
(XLSX)

**S107 Table. GISAID acknowledgement table, containing list of sequences used, for RNAdescent analysis of NP sequences from H7N9 viruses originating from avian hosts.**
(XLSX)

**S108 Table. GISAID acknowledgement table, containing list of sequences used, for RNAdescent analysis of NA sequences from H7N9 viruses originating from avian hosts.**
(XLSX)

**S109 Table. GISAID acknowledgement table, containing list of sequences used, for RNA-descent analysis of M1 sequences from H7N9 viruses originating from avian hosts.**
(XLSX)

**S110 Table. GISAID acknowledgement table, containing list of sequences used, for RNA-descent analysis of M2 sequences from H7N9 viruses originating from avian hosts.**
(XLSX)

**S111 Table. GISAID acknowledgement table, containing list of sequences used, for RNA-descent analysis of NS1 sequences from H7N9 viruses originating from avian hosts.**
(XLSX)

**S112 Table. GISAID acknowledgement table, containing list of sequences used, for RNA-descent analysis of NS2 sequences from H7N9 viruses originating from avian hosts.**
(XLSX)

**S1 Code. Mathematica notebook containing analysis output from H1N1pdm09 viruses originating from human hosts.** The notebook contains, for each gene analysed:

1. A histogram of amino acid lengths of analysed sequences;

2. A plot of Shannon information versus the score (raw codon variability values) at each codon in the gene;

3. An example sequence from the gene with nucleotide loci found to be constrained highlighted (in orange if found to be constrained without needing to adjust for potentially interfering regions, in blue if only found after adjusting for potentially interfering regions);

4. A codon-level report of nucleotide usage (with observed and expected values for usage, mean distance, normalised ratio, weighted normalised ratio, weighted normalised ratio after all ratios across the gene are scaled to a mean of zero and variance of one, and ranking of the codon after weighting and scaling, with the scaled values highlighted in orange or blue if in constrained regions, similarly to the example sequence highlighting);

5. A plot showing the proportion of sequences at each codon where the observed amino acid produced is the most common one;

6. Plots showing the weighted constraint scores for each codon for the raw (unranked) and ranked analyses, plus plots of cumulative sums of these scores (again with codons in constrained regions highlighted orange or blue, similarly to the example sequence highlighting, and with codons with zero weight plotted with open triangles and with paler colours).
The zip file contains a copy of the notebook in Mathematica format and in pdf.
(ZIP)

**S2 Code. Mathematica notebook containing analysis output from H1N2 viruses originating from swine hosts.** The content of the notebook follows the same pattern as that in S1 Code.
(ZIP)

**S3 Code. Mathematica notebook containing analysis output from H3N2 viruses originating from human hosts.** The content of the notebook follows the same pattern as that in S1 Code.
(ZIP)

**S4 Code. Mathematica notebook containing analysis output from H5N1 viruses originating from avian hosts.** The content of the notebook follows the same pattern as that in S1

Code.
(ZIP)

**S5 Code. Mathematica notebook containing analysis output from H5N8 viruses originating from avian hosts.** The content of the notebook follows the same pattern as that in S1 Code.
(ZIP)

**S6 Code. Mathematica notebook containing analysis output from H7N7 viruses originating from avian hosts.** The content of the notebook follows the same pattern as that in S1 Code.
(ZIP)

**S7 Code. Mathematica notebook containing analysis output from H7N9 viruses originating from human hosts.** The content of the notebook follows the same pattern as that in S1 Code.
(ZIP)

**S8 Code. Mathematica notebook containing analysis output from H7N9 viruses originating from avian hosts.** The content of the notebook follows the same pattern as that in S1 Code.
(ZIP)

**S1 Fig. RNAalifold output from alignments of constrained regions in H1N1pdm09 viruses originating from human hosts, as predicted by RNAdescent.** Filename convention within the zip file is as follows: HA/NA type, then host type, then whether the region was identified through analysis of raw (unranked) data, ranked data, or both, then the gene name, then the nucleotide location within the analysed alignment of the gene name, then the nucleotide location within the reference sequences (corresponding to locations listed in S2 Table and S10 Table), then a note if the analysis was performed using only one example of each distinct sequence, then a note if the fold uses the reverse complement of the cRNA (i.e. the vRNA), rather than the cRNA. All folds have been generated using alignments with loci where the consensus nucleotide is a gap removed. Base pairs are highlighted in deep/mid/light red when all/all but one/all but two sequences are capable of forming the pairs shown. Base pairs are highlighted in deep/mid/light yellow when all/all but one/all but two sequences are capable of forming the pair shown or one other pair (including GU pairs). Base pairs are highlighted in deep/mid/light green when all/all but one/all but two sequences are capable of forming the pair shown or one of two other pairs (including GU pairs). RNAalifold was used with input options disallowing lonely pairs, allowing G-quadruplexes, and with the ribosum scoring matrix enabled.
(ZIP)

**S2 Fig. RNAalifold output from alignments of constrained regions in H1N2 viruses originating from swine hosts, as predicted by RNAdescent.** See the caption for S1 Fig for a description of the filename convention (save that the corresponding nucleotide locations in reference sequences are listed in S3 and S11 Tables), and an explanation of the RNAalifold options used and output.
(ZIP)

**S3 Fig. RNAalifold output from alignments of constrained regions in H3N2 viruses originating from human hosts, as predicted by RNAdescent.** See the caption for S1 Fig for a description of the filename convention (save that the corresponding nucleotide locations in

reference sequences are listed in S4 and S12 Tables), and an explanation of the RNAalifold options used and output.
(ZIP)

**S4 Fig. RNAalifold output from alignments of constrained regions in H5N1 viruses originating from avian hosts, as predicted by RNAdescent.** See the caption for S1 Fig for a description of the filename convention (save that the corresponding nucleotide locations in reference sequences are listed in S5 and S13 Tables), and an explanation of the RNAalifold options used and output (save that for these avian-origin viruses the folding temperature was set to 41˚C).
(ZIP)

**S5 Fig. RNAalifold output from alignments of constrained regions in H5N8 viruses originating from avian hosts, as predicted by RNAdescent.** See the caption for S1 Fig for a description of the filename convention (save that the corresponding nucleotide locations in reference sequences are listed in S6 and S14 Tables), and an explanation of the RNAalifold options used and output (save that for these avian-origin viruses the folding temperature was set to 41˚C).
(ZIP)

**S6 Fig. RNAalifold output from alignments of constrained regions in H7N7 viruses originating from avian hosts, as predicted by RNAdescent.** See the caption for S1 Fig for a description of the filename convention (save that the corresponding nucleotide locations in reference sequences are listed in S7 and S15 Tables), and an explanation of the RNAalifold options used and output (save that for these avian-origin viruses the folding temperature was set to 41˚C).
(ZIP)

**S7 Fig. RNAalifold output from alignments of constrained regions in H7N9 viruses originating from human hosts, as predicted by RNAdescent.** See the caption for S1 Fig for a description of the filename convention (save that the corresponding nucleotide locations in reference sequences are listed in S8 and S16 Tables), and an explanation of the RNAalifold options used and output.
(ZIP)

**S8 Fig. RNAalifold output from alignments of constrained regions in H7N9 viruses originating from avian hosts, as predicted by RNAdescent.** See the caption for S1 Fig for a description of the filename convention (save that the corresponding nucleotide locations in reference sequences are listed in S9 and S17 Tables), and an explanation of the RNAalifold options used and output (save that for these avian-origin viruses the folding temperature was set to 41˚C). As some of the constrained regions predicted in the analysis of the M2 gene using raw (unranked) codon variability values are very close together, alignments of longer regions containing more than one conserved region are also included.
(ZIP)

## Acknowledgments

We gratefully acknowledge all data contributors, i.e., the Authors and their Originating laboratories responsible for obtaining the specimens, and their Submitting laboratories for generating the genetic sequence and metadata and sharing via the GISAID Initiative, on which this research is based.

JPS is grateful to Charlotte Houldcroft for critical reading of an earlier version of this manuscript.

We are grateful to the Cambridge Mathematics Placements scheme for facilitating this study.

## Author Contributions

**Conceptualization:** Jordan P. Skittrall.

**Data curation:** Emma Beniston, Jordan P. Skittrall.

**Formal analysis:** Emma Beniston, Jordan P. Skittrall.

**Investigation:** Emma Beniston, Jordan P. Skittrall.

**Methodology:** Jordan P. Skittrall.

**Supervision:** Jordan P. Skittrall.

**Visualization:** Jordan P. Skittrall.

**Writing – original draft:** Jordan P. Skittrall.

**Writing – review & editing:** Emma Beniston, Jordan P. Skittrall.

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
