## [Decision Letter · Decision Letter 0]

19 Dec 2023

Dear Dr Skittrall,

Thank you very much for submitting your manuscript "Locations and structures of influenza A packaging-associated signals and other functional elements via an *in silico* pipeline for predicting conserved features in RNA viruses" for consideration at PLOS Computational Biology.

As with all papers reviewed by the journal, your manuscript was reviewed by members of the editorial board and by several independent reviewers. In light of the reviews (below this email), we would like to invite the resubmission of a significantly-revised version that takes into account the reviewers' comments.

The reviewers raised several critical issues, including concerns about what exactly is new in this work, the problem with the link to secondary structure, the actual functional relevance (which was problematic), the comparison and validation with recent in vivo structure probing data, the inclusion of long-range contacts, and alternative structures. A substantial revision is expected to carefully address these issues should you decide to resubmit the manuscript.

We cannot make any decision about publication until we have seen the revised manuscript and your response to the reviewers' comments. Your revised manuscript is also likely to be sent to reviewers for further evaluation.

Sincerely,

Shi-Jie Chen

Academic Editor

PLOS Computational Biology

William Noble

Section Editor

PLOS Computational Biology

Reviewer's Responses to Questions

**Comments to the Authors:**

Reviewer #1: This manuscript reports on the results of a sequence and structural analysis of influenza A virus RNAs. The authors utilize a novel approach for estimating constraints on codon evolution and apply it to various strains of IAV to deduce regions with significant constraint. Using this they extract subregions and alignments for secondary structure prediction via RNAalifold. The authors use codon level constraints to highlight known functional sites, such as packaging signals, and post-transcriptional regulatory elements that include structured RNAs previously described, then go on to discuss potentially novel regions that arise from their analysis. My main concern with this paper is that it revisits a target that has been previously studied using similar approaches (identification of constrained codon usage, secondary structure prediction, and comparative sequence analysis) and was subsequently extensively studied using multiple in vitro and in cellulo chemical structure probing approaches and covariation analysis. To add value to this field, this work has to improve or expand on this deep body of previous work, but the manuscript, as presented, only partially does this.

The major contribution of this current work, in my opinion, is the ability to home in on codon level constraints to better identify local regions of constraint that may be functional (beyond coding for proteins). It is difficult though to determine what exactly we learn that is new, due to extensive digressions away from this main finding. In particular, the attempt to link this to secondary structure was problematic, as RNAalifold is highly sensitive to the sequences aligned and the extent of sequence fragments analyzed. The criteria the authors used - organizing sequences by antigenic subtype and clustered constrained codon sites - almost guarantees the highly diverse (in structure and levels of conservation) structure models presented in each of their seventeen main text figures: e.g. the variable and poorly conserved structures presented in Fig. 4. I don’t think there is much value in presenting these models as main text figures, as the predicted differences between strains and RNAdescent hits are arising from biases in the “covariation” score in RNAalifold (arising from sequences used) and the potential alternative pairing from having more or less sequence to fold - and are not likely to represent actual functionally significant conformational differences between strains.

My recommendation is for a significant revision of the manuscript to emphasize the major novel contributions of their analysis, which are the RNAdescent results. For RNA structural analyses, I think the RNAalifold results could perhaps be added as supplemental data, if presented at all (primarily as a point of comparison to existing models). For the main text it would be valuable to focus on interpreting previous structure analyses through the lens of their novel results: e.g. as they did when revisiting the pseudoknot model presented by Priore et. al. Most IAV sequence fragments have been probed and/or modeled at this point. Please note some notable example publications below:

https://academic.oup.com/nar/article/47/13/7003/5485530

https://doi.org/10.1016/j.csbj.2023.10.036

https://www.nature.com/articles/s41591-022-01908-x

https://doi.org/10.1016/j.celrep.2020.107823

https://www.nature.com/articles/s41598-021-03767-x

If there is a compelling reason to perform additional structural analyses: e.g. if strain specific conformations are strongly suspected or a region was missed in previous work, a more careful approach for modeling should be used. RNAalifold results could serve as a good starting point, but artifacts arising from the length and the choice of sequences need to be considered when determining final models to present. Additionally, validation of RNAalifold predictions using a tool like R-Scape could be especially valuable here, where sites of significant covariation could overlap constrained codons.

Minor Comments:

Rather than use the term “folds” it may be better to refer to “secondary structures”, “secondary structure models/predictions”, etc. “Fold” seem to imply alternate conformations.

Throughout the manuscript “conserved codons” are discussed but I think “constrained codons” would be a better term to describe what they mean.

Little typos like spelling errors and missing articles: e.g.

Line 5: “hospitalizatons“ should be “hospitalizations”

Line 88: and then THE analysis WAS performed

Line 96: we performed THE analysis

A few rounds of additional review would be good.

Reviewer #2: Beniston et. al. utilized RNAdescent algorithms to analyze the influenza A virus (IAV) sequences from the GISAID database. Their studies identified highly conserved regions, and predicted conserved RNA secondary structures that are related with packaging and other critical functions. IAV poses a significant threat to human public health by causing seasonal outbreaks and potential pandemics, while also inflicting economic losses in poultry industries due to the virus's ability to infect avian species, resulting in widespread poultry losses. The complexity of IAV subtypes across various hosts has made IAV RNA structure-function studies challenging. The authors employed computational tools to assess sequence conservation within and among subtypes, predicting consensus secondary structures for RNAs in these conserved regions. Many identified conserved structural regions are implicated in virus packaging, translation, and splicing. This systematic analysis provides a robust tool for examining previous observed experimental data and guiding the future experiment design. The manuscript presents an extensive analysis of RNA structural motifs in each segment RNA, providing valuable structural information to explain previous experimental observations.

I’d recommend the authors to consult recent in vivo RNA structure probing papers to validate their approaches and models. The in vivo H1N1 mRNA (cRNA) and vRNA structures have been probed by SHAPE and secondary structure models have been presented (E.g. Simon et. al. 2019 PMC6648356); Mirska et. al. 2023 PMC10153785). The authors should validate their structural models with these experimental data.

In terms of packaging signals, the authors reviewed a few packaging papers and predicted how the mutations affect the RNA secondary structures and packaging based on their models. IAV RNA packaging and reassortment are complicate, and RNA long-range interactions among various vRNA segments have been observed by several groups (Dadonaite et. al. 2019, PMC7191640 Sage. et. al. 2020; PMC7372595.) It would be very helpful to know that whether these residues in the observed inter-segment interactions are conserved. Are they predicted to form local structures by RNAalifold or they are single stranded and thus can readily form inter-segment long-range interactions?

The secondary structures of RNA sequences in the conserved regions were analyzed using RNAalifold, which take co-variation into consideration when predicting RNA secondary structures. However, only one consensus structures are generated for each set of sequences. This raises the concern that how the software handles RNAs that exist in equilibrium of multiple conformations or switching between conformations for function regulation. PA-X frameshift stimulator structures shown in Fig 15 varies among different subtypes. I wonder if the RNA structure prediction algorithm allows searching a few alternative structures with comparable free energies. If so, a consensus structure maybe observed among IAV subtypes.

IAV RNAs are coated with nucleoprotein (NP), and how the NP binding affects vRNA and cRNA secondary structure is not discussed in the manuscript. The current approaches in the manuscript may not be able to address this issue, but it should be included in the discussion.

the Author Summary, the authors stated their work “allows design and evaluation of vaccine candidates to ensure attenuated viruses remain able to stimulate the immune system”. I feel this is overstated as there was no such description in the results section. It was not clear to me in the Discussion (lines 669-671) “we highlighted where appropriate mutation may result in attenuated vaccine strains…”.

Fig. 16 shows a stem-loop with high number of G-C pairs in the NS intron, by manually inspection of the sequence. Did RNAalifold predict such structure? What is the rationale to switch from RNAalifold in this case?

A stem-loop motif was observed in the HA of H7N9 that has potential to transit to high-pathogenicity avian influenza viruses. Can the authors please comment on their findings on HA of H5 subtypes? H5 HA also has been shown to be able to evolve from low pathogenicity to high pathogenicity viruses.

Fig. 5 is missing.

Some of the RNA secondary structure figures are difficult to read. It will be helpful to include some labels, such as the 5’-. 3’-, and some residue numbers.

**Have the authors made all data and (if applicable) computational code underlying the findings in their manuscript fully available?**

Reviewer #1: Yes

Reviewer #2: Yes

PLOS authors have the option to publish the peer review history of their article (what does this mean?). If published, this will include your full peer review and any attached files.

Reviewer #1: No

Reviewer #2: No
---

## [Decision Letter · Decision Letter 1]

18 Mar 2024

Dear Dr Skittrall,

We are pleased to inform you that your manuscript 'Locations and structures of influenza A packaging-associated signals and other functional elements via an *in silico* pipeline for predicting constrained features in RNA viruses' has been provisionally accepted for publication in PLOS Computational Biology.

Best regards,

Shi-Jie Chen

Academic Editor

PLOS Computational Biology

William Noble

Section Editor

PLOS Computational Biology

Reviewer's Responses to Questions

**Comments to the Authors:**

Reviewer #1: Authors have made across the board improvements to the manuscript and have addressed my comments and concerns.

Reviewer #2: The authors have addressed my questions/concerns by rearranging the data and expanding the discussion section. Recommend for publication.

**Have the authors made all data and (if applicable) computational code underlying the findings in their manuscript fully available?**

Reviewer #1: Yes

Reviewer #2: Yes

PLOS authors have the option to publish the peer review history of their article (what does this mean?). If published, this will include your full peer review and any attached files.

Reviewer #1: No

Reviewer #2: No

---

## [Editor Report · Acceptance letter]

2 Apr 2024

PCOMPBIOL-D-23-01793R1 

Locations and structures of influenza A virus packaging-associated signals and other functional elements via an in silico pipeline for predicting constrained features in RNA viruses

Dear Dr Skittrall,

I am pleased to inform you that your manuscript has been formally accepted for publication in PLOS Computational Biology. Your manuscript is now with our production department and you will be notified of the publication date in due course.

With kind regards,

Anita Estes
